# REPOSITIONING THE SUBJECT WITHIN IMAGE

## ABSTRACT

Current image manipulation primarily centers on static manipulation, such as replacing specific regions within an image or altering its overall style. In this paper, we introduce an innovative dynamic manipulation task, subject repositioning. This task involves relocating a user-specified subject to a desired position while preserving the image's fidelity. Our research reveals that the fundamental sub-tasks of subject repositioning, which include filling the void left by the repositioned subject, reconstructing obscured portions of the subject and blending the subject to be consistent with surrounding areas, can be effectively reformulated as a unified, prompt-guided inpainting task. Consequently, we can employ a single diffusion generative model to address these sub-tasks using various task prompts learned through our proposed task inversion technique. Additionally, we integrate pre-processing and post-processing techniques to further enhance the quality of subject repositioning. These elements together form our SEgment-gEnerate-and-bLEnd (SEELE) framework. To assess SEELE's effectiveness in subject repositioning, we assemble a real-world subject repositioning dataset called ReS. Our results on ReS demonstrate the quality of repositioned image generation.

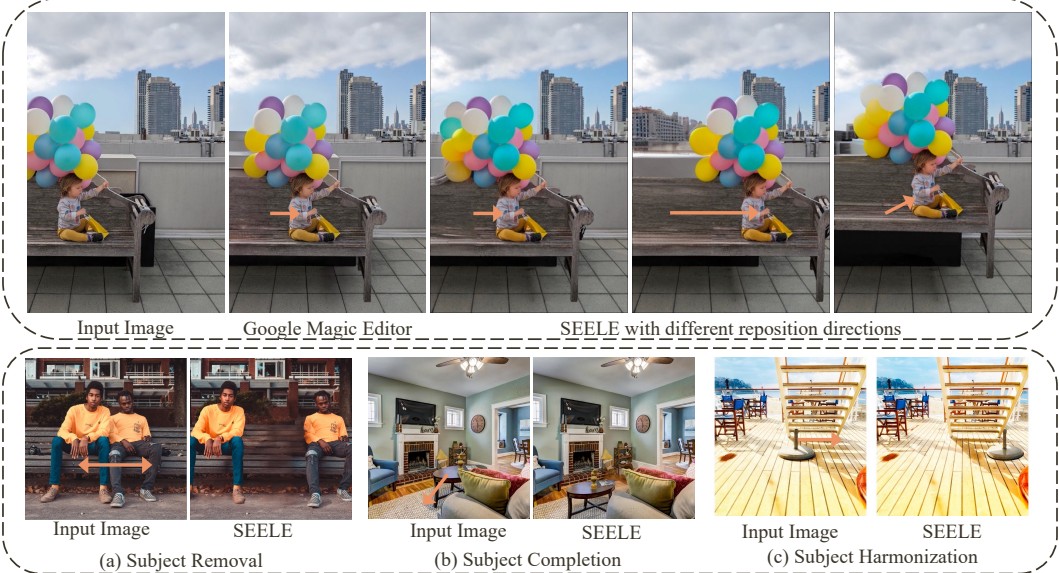

Figure 1: Subject repositioning aims to relocate a user-specified subject within a single image. In the comparison **above**, we evaluate the subject repositioning results achieved by our SEELE model in comparison to Google Magic Editor. We obtained Google's results from its introductory webpage. **Below** are illustrated generative sub-tasks encompassed by subject repositioning: i) It must fill the void created when moving the subject to maintain consistency and avoid generating new, random subjects. ii) Completing the occluded portions of the moved subject is necessary. iii) The appearance of repositioned subject should blend with the surrounding areas. SEELE effectively addresses the generative sub-tasks within a unified prompt-guided inpainting task, all powered by a single diffusion generative model. While these results illustrate the sub-tasks addressed by SEELE, the comprehensive outcomes of executing SEELE are depicted in Figure 13 in the appendix.

# 1 INTRODUCTION

In May 2023, Google Photos introduced a groundbreaking AI editing feature allowing users to reposition subjects within their images[1]. Unfortunately, a lack of accompanying technical documentation leaves the inner workings of this feature largely unexplored. Prior to the deep learning era, Iizuka et al. (2014) explored a similar problem of object repositioning with user inputs of ground regions, bounding boxes of objects, and shadow regions to aid the understanding of the image. As deep learning has rapidly advanced, the potential to substitute many user actions with learning models as well as an advanced understanding of images has emerged, necessitating a comprehensive reassessment of the subject repositioning problem through the lens of potent deep learning models. The primary objective of this paper is to introduce an inventive framework capable of achieving performance on par with or surpassing Google Photos' latest AI feature for repositioning subjects within images.

From an academic standpoint, it's evident that this feature falls within the domain of image manipulation (Gatys et al., 2016; Isola et al., 2017; Zhu et al., 2017; Wang et al., 2018; El-Nouby et al., 2019; Fu et al., 2020; Zhang et al., 2021). This area has seen a surge in interest in recent years, primarily due to the advancement of large-scale generative models. These generative models encompass a range of techniques, including generative adversarial models (Goodfellow et al., 2014), variational autoencoders (Kingma & Welling, 2014), auto-regressive models (Vaswani et al., 2017), and notably, diffusion models (Sohl-Dickstein et al., 2015). As both the model architectures and training datasets continue to expand, these generative models exhibit remarkable capabilities in image manipulation (Rombach et al., 2022; Kawar et al., 2022; Chang et al., 2023).

However, it is important to note that current image manipulation approaches primarily emphasize what can be described as "static" alterations. These methods are designed to modify specific regions of an image, often guided by various cues such as natural language, sketches, strokes, or layouts (El-Nouby et al., 2019; Zhang et al., 2021; Fu et al., 2020). Another dimension of manipulation revolves around the transformation of an image's overall style, encompassing tasks like converting real photographs into anime-style pictures, paintings, or mimicking the unique aesthetics of certain films (Chen et al., 2018; Wang et al., 2018; Jiang et al., 2021). Some approaches have even extended these manipulation techniques to the domain of videos (Kim et al., 2019; Xu et al., 2019; Fu et al., 2022), where the objective is to dynamically manipulate style or subjects over time. In contrast, the concept of subject repositioning delves into the dynamic manipulation of a single image, with a specific focus on relocating selected subject while keeping the rest of the image unchanged.

As text-to-image diffusion models (Nichol et al., 2022; Ho et al., 2022; Saharia et al., 2022; Ramesh et al., 2022; Rombach et al., 2022) emerge as one of the most potent generative models today, adapting them for subject repositioning presents an intriguing opportunity. Nevertheless, a significant challenge lies in finding suitable text prompts for this task, as text-to-image diffusion models are typically trained using image caption prompts rather than task-specific instructions. Moreover, the best text prompts are often image-dependent and are hard to generalize to other images, making them impractical for real-world applications that prioritize user-friendliness and minimal user effort. On the other hand, while specialized models have been developed to address specific aspects of subject repositioning, such as local inpainting (Zeng et al., 2020; Zhao et al., 2021; Li et al., 2022; Suvorov et al., 2022; Dong et al., 2022), subject completion (Zhan et al., 2020), and local harmonization (Xu et al., 2017; Zhang et al., 2020; Tsai et al., 2017), our study poses an intriguing question: "*Can we achieve all these sub-tasks using a single generative model?*"

Broadly, we can deconstruct this multifaceted task into several distinct sub-tasks. We roughly categorize these sub-tasks into non-generative and generative tasks. The non-generative sub-tasks involve activities like segmenting user-specified subjects and estimating occlusion relationships between subjects. In this paper, we primarily concentrate on the generative sub-tasks, while addressing the non-generative aspects using pre-trained models.

The generative sub-tasks essential for subject repositioning encompass the following key elements: i) **Subject removal**: After the subject is repositioned, a void is left behind. The generative model's task is to consistently fill this void using nearby background while avoiding the introduction of new elements. ii) **Subject completion**: When the repositioned subject is partially obscured, the

---

[1] https://blog.google/products/photos/google-photos-magic-editor-pixel-io-2023/

generative model must complete the subject to maintain its integrity. iii) **Subject harmonization**: The appearance of repositioned subject should seamlessly blend with the surrounding areas.

While all these sub-tasks take as inputs an image for manipulation and a mask indicating the region to manipulate, they demand distinct generative capabilities. Furthermore, it is hard to transform these task instructions into caption-style prompts for frozen text-to-image diffusion models.

Fortunately, the embedding space of text prompts used in diffusion models is much more versatile than merely representing captions. Textual inversion (Gal et al., 2022) has revealed that we can learn to represent user-specified concepts, including textual and stylistic information that is challenging to convey through language, within the embedding space of text prompts. Additionally, prompt tuning (Lester et al., 2021; Liu et al., 2021a) has been effectively employed in transformers to adapt to specific domains, inspiring us to apply textual inversion at the task level. These approaches inspire us to learn latent embeddings in the text conditions to represent specific task instructions that the diffusion model should follow. With this task-level inversion design, we can adapt diffusion models to various tasks by simply modifying the task-level "text" prompts.

To formally address the problem of subject repositioning, we propose the SEgment-gEnerate-and-bLEnd (SEELE) framework. SEELE tackles the subject repositioning with a pre-processing, manipulation, post-processing pipeline. i) In the pre-processing stage, SEELE employs SAM (Kirillov et al., 2023) to input user-specified points, bounding boxes, or text prompts to segment the subject for repositioning. With the user-specified moving direction, SEELE moves the subject and places it following the accurate occlusion relationship between subjects. ii) In the manipulation stage, SEELE addresses subject removal and subject completion using a single pre-trained diffusion model guided by learned task prompts. iii) In the post-processing stage, SEELE harmonizes the repositioned subject to ensure it blends seamlessly with adjacent regions.

To evaluate subject repositioning algorithms, we have assembled a real-world subject repositioning dataset called ReS. This dataset consists of 100 real image pairs featuring a repositioned subject. The images were collected in diverse scenes and at different times to enhance diversity. We annotated the mask of the repositioned subject using SAM and manual refinement. We estimated the moving direction based on the center point of masks in the paired image. We also provide amodal masks for occluded subjects. To the best of our knowledge, this is the first dataset for subject repositioning, and we hope it will serve as a valuable benchmark evaluation dataset.

Our contributions are summarized as follows:

- The paper delineates the task of subject repositioning as a specialized image manipulation challenge, breaking it down into several distinct sub-tasks, each of which presents unique challenges and necessitates specific learning model requirements.
- The paper proposes the SEgment-gEnerate-and-bLEnd (SEELE) framework, which addresses multiple generative sub-tasks in subject repositioning using a single diffusion model.
- The paper explores an innovative task inversion technique, demonstrating that we can re-formulate the text-conditions to represent task instructions. This exploration opens up new possibilities for adapting diffusion models to specific tasks.
- The paper curates the ReS dataset, a real-world collection of image pairs featuring repositioned subjects. ReS serves as a valuable benchmark for evaluating subject repositioning algorithms.

## 2 SUBJECT REPOSITIONING

### 2.1 TASK DEFINITION AND CHALLENGES

Subject repositioning involves moving the user-specified subject within an image. This seemingly simple task is actually quite challenging, requiring coordination of multiple sub-tasks.

**User inputs**. Subject repositioning focuses on a single image. As an interactive approach, subject repositioning follows user-intention to identify subject, move to the desired location, complete the subject and address disparities of the repositioned subject. Particularly, the user identify the interested subject via pointing, bounding box, or a text prompt as inputs to the system for identifying the subject. Then, the user provides the desired repositioning location via dragging or providing repositioning direction. The system further requires the user to indicate the

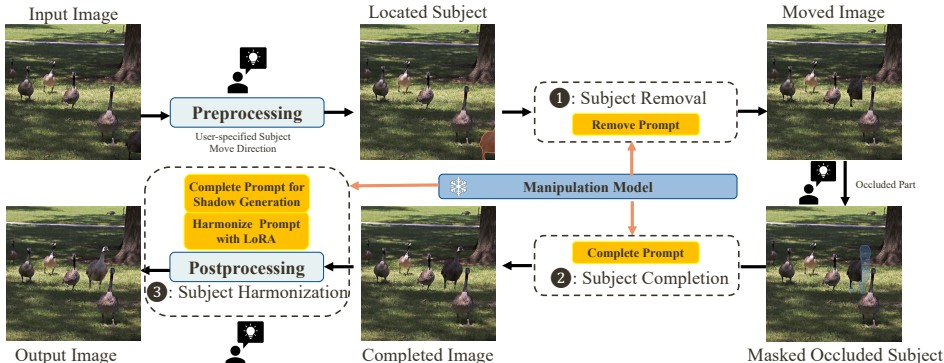

Figure 2: SEELE employs a pre-processing, manipulation, and post-processing pipeline for subject repositioning. During the pre-processing phase, SEELE identifies the subject using the segmentation model, guided by user-provided conditions, and maintains the occlusion relationships between subjects intact. In the manipulation stage, SEELE manipulates the image to fill in any left gaps. Furthermore, SEELE rectifies the obscured subject with user-specified incomplete masks. In the post-processing phase, SEELE addresses any disparities between the repositioned subject and its new surroundings.

occluded part of the subject for completion, as well as whether running particular postprocessing algorithms for minimizing visual differences. An illustration of user inputs is shown in Figure 3.

To tackle this task, we introduce the SEgment-gEnerate-and-bLEnd (SEELE) framework, shown in Figure 2. Specifically, SEELE breaks down the task into three stages: preprocessing, manipulation, and post-processing stages.

i) The *preprocessing* addresses how to precisely locate the specified subject with minimal user input, considering that the subject may be a single object, part of an object, or a group of objects identified by the user's intention; reposition the identified subject to the desired location; and also identify occlusion relationships to maintain geometric consistency. Additionally, adjusting the subject's size might be necessary to maintain the perspective relationship within the overall composition.

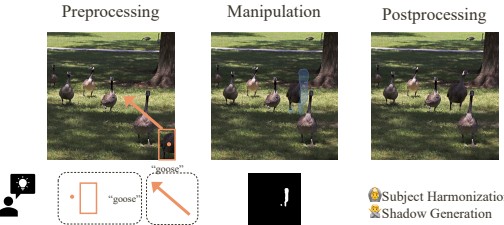

Figure 3: Illustration of user inputs in each stage.

ii) The *manipulation* stage deals with the main tasks of creating new elements in subject repositioning to enhance the image. In particular, this stage includes the subject removal step, which fills the empty space on the left void of the repositioned subject. Additionally, the subject completion step involves reconstructing any obscured parts to ensure the subject is fully formed.

iii) The *postprocessing* stage focuses on minimizing visual differences between the repositioned subject and its new surroundings. This involves fixing inconsistencies in both appearance and geometry, including blending unnatural boundaries, aligning illumination statistics, and, at times, creating realistic shadows for added realism.

In the next sections, we will start by going over the SEELE pipeline in Sec. 2.2. Particularly we explain the task inversion in Sec. 2.3 to address generative sub-tasks. In Sec. 2.4, we show how to train different manipulation sub-tasks using the task inversion technique while keeping the diffusion model unchanged. Finally, we provide a detailed introduction to the curated ReS dataset in Sec. 2.5.

## 2.2 SEELE

As mentioned above, SEELE consists of three stages. The preprocessing stage usually involves non-generative tasks, while the manipulation and postprocessing stages require generative capabilities. In SEELE, we employ a unified diffusion model for all generative sub-tasks and use pre-trained models for non-generative sub-tasks. We give the details of each stage in the following.

**Pre-processing**. For point and bounding box inputs for identify subject, we utilize the SAM (Kirillov et al., 2023) for user interaction and employ SAM-HQ (Ke et al., 2023) to enhance the quality of segmenting subjects with intricate structures. To enable text inputs, we follow SeMani (Wang et al., 2023) to indirectly implement a text-guided SAM mode. Specifically, we first employ SAM to segment the entire image into distinct subjects. Subsequently, we compare each subject with the input text to identify the most similar one using the mask-adapted CLIP model (Liang et al., 2022).

After identifying the subject, SEELE follows user intuition to reposition the subject to the desired location, and then mask the original subject as void for re-paint in the manipulation stage.

Our SEELE handles the potential occlusion between the moved subject and other elements in the image. If there are other subjects present at the desired location, SEELE employs the monocular depth estimation algorithm MiDaS (Ranftl et al., 2020) to discern occlusion relationships between subjects. SEELE will then appropriately mask the occluded portions of the subject if the user wants to preserve these occlusion relationships. MiDaS is also used to estimate the perspective relationships among subjects and resize the subject accordingly to maintain geometric consistency. For subjects with ambiguous boundaries, SEELE incorporates the ViTMatte matting algorithm (Yao et al., 2023) for better compositing with the surrounding areas.

**Manipulation**. In this stage, SEELE deals with the primary tasks of manipulating subjects by repositioning them. As illustrated in Figure 2, it has the steps of subject removal and subject completion. Critically, such two steps can be effectively solved by a single generative model, as the masked region of both steps should be filled in to match the surrounding areas. However, these two sub-tasks require different information and types of masks. Particularly, for subject removal, a *non-semantic* inpainting is applied uniformly from the unmasked regions, using a typical object-shaped mask. This often falsely results in the creation of new, random subjects within the holes. On the other hand, subject completion involves *semantic-rich* inpainting and aims to incorporate the majority of the masked region as part of the subject. Critically, to adapt the same diffusion model to the different generation directions needed for the above sub-tasks, we propose the task inversion technique in SEELE. This technique guides the diffusion model according to specific task instructions. Thus, with the learned *remove-prompt* and *complete-prompt*, SEELE combines subject removel and subject completion into a single generative model.

**Post-processing**. In the final stage, SEELE harmoniously blends the repositioned subject with its surroundings by tackling two challenges below.

i) *Local harmonization* ensures natural appearance in boundary and lighting statistics. SEELE confines this process to the relocated subject to avoid affecting other image parts. It takes the image and a mask indicating the subject's repositioning as inputs. However, the stable diffusion model is initially trained to generate new concepts within the masked region, conflicting with our goal of only ensuring consistency in the masked region and its surroundings. To address this, SEELE adapts the model by learning a *harmonize-prompt* and using the LoRA adapter to guide masked regions. This local harmonization can also be integrated into the same diffusion model used in the manipulation stage with our newly proposed design.

ii) *Shadow generation* aims to create realistic shadows for repositioned subjects, enhancing the realism. Generating high-fidelity shadows in high-resolution images of diverse subjects remains challenging. SEELE uses the stable diffusion model for shadow generation, addressing two scenarios: (1) If the subject already has shadows, we use *complete-prompt* for subject completion to extend the shadows. (2) For subjects without shadows, we generate a preliminary shadow based on user-specified masks. This task then transforms into a local harmonization process for realistic shadow generation, utilizing *harmonize-prompt* with LoRA adapter Hu et al. (2021).

## 2.3 TASK INVERSION

Generative sub-tasks in subject repositioning input the image and mask with unqiue approach:

- Subjecr removal fills the void without creating new subjects.
- Subject completion completes the primary subject within the masked region.
- Subject harmonization ensures consistency without introducing new elements.

These requirements lead to different generation paths. In contrast, our goal is to enhance text-to-image diffusion inpainting models for image manipulation guided by high-level task instructions.

To address this, we introduce task inversion, training prompts to guide the diffusion model while keeping the backbone fixed. Instead of traditional text prompts, we utilize the adaptable representations acting as instruction prompts, such as "complete the subject". Consequently, task inversion allows the smooth integration of different generative sub-tasks for subject repositioning using stable diffusion. This integration happens without the need for introducing new generative models or adding extensive modules or parameters, highlighting the plug-and-play nature of task inversion.

Task inversion adheres to the original training objectives of diffusion models. Specifically, denote the training image as $x$, the local mask as $m$, the learnable task prompt as $z$, the conditioning model $c(\cdot)$ to map the learnable prompt. Our objective is

$$\mathcal{L} := \mathbb{E}_{\varepsilon \sim \mathcal{N}(0,1), t \sim \mathcal{U}(0,1)}[\|\varepsilon - \varepsilon_\theta([x_t, m, x \odot (1 - m)], t, c(z)\|_F^2], \quad (1)$$

where $\varepsilon$ is the random noise; $\varepsilon_\theta$ is the diffusion model, $t$ is the normalized noise-level; $x_t$ is the noised image, $\odot$ is element-wise multiplication; and $\|\cdot\|_F$ is the Frobenius norm. When training with Eq. (1), the conditioning model $c$ and the diffusion model $\varepsilon_\theta$ is frozen, while the embedding $z$ is the only learnable parameters.

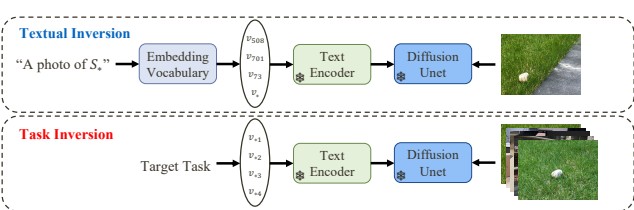

Figure 4: Task inversion expands upon textual inversion, addressing different objectives, employing distinct training methodologies, and serving various tasks.

Our task inversion is a distinctive approach, influenced by various existing works but with clear differences. Specifically, traditional text-to-image diffusion models are trained on pairs where the text describes the image, such as LAION-5B (Schuhmann et al., 2022). However, the instruction prompt mentioned for our task inversion goes beyond the training data's scope, potentially affecting the desired generation results in practice. Furthermore, recent advancements in textual inversion (Gal et al., 2022) emphasize the potential to comprehend user-specified concepts within the embedding space. In contrast, prompt tuning (Lester et al., 2021; Liu et al., 2021a) enhances adaptation to specific domains by introducing learnable tokens to the inputs. Similarly, adversarial reprogramming (Elsayed et al., 2018) trains a pre-existing model to perform a novel task. Unlike textual inversion, which trains a few tokens for visual understanding, our task prompt includes the entire task instruction. We don't depend on text inputs to guide the diffusion model; instead, we use all tokens for learning. See Figure 4 for the distinction.

## 2.4 LEARNING TASK INVERSION

Existing text-to-image diffusion inpainting model is trained with randomly generated masks to generalize in diverse scenarios. In contrast, task inversion involves creating task-specific masks during training, allowing the model to learn specialized task prompts.

i) *Generating masks for subject removal*: In subject repositioning, the mask for the left void mirrors the subject's shape, but our goal isn't to generate the subject within the mask. To create training data for this scenario, for each image, we randomly choose a subject and its mask. Next, we move the mask, as shown by the girl's mask in the center of Figure 5. This results in an image where the masked region includes random portions unrelated to the mask's shape. This serves as the target for subject removal, with the mask indicating the original subject location.

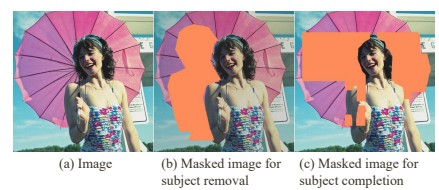

(a) Image    (b) Masked image for subject removal    (c) Masked image for subject completion

Figure 5: Generating Masks to train subject removal and subject completion.

ii) *Generating masks for subject completion*: In this phase, SEELE addresses scenarios where the subject is partially obscured, with the goal of effectively completing the subject. To integrate this

prior information into the task prompt, we generate training data as follows: for each image, we randomly select a subject and extract its mask. Then, we randomly choose a continuous portion of the mask as the input mask. Since user-specified masks are typically imprecise, we introduce random dilation to include adjacent regions within the mask. As illustrated by the umbrella mask on the right side of Figure 5, such a mask serves as an estimate for the mask used in subject completion.

**Learning subject harmonization**. In SEELE, we refine subject harmonization by altering the target of diffusion model. This change replaces the masked image condition with the original inharmonious image in Eq. (1). Task inversion mainly influences the cross-attention between the task condition and images. Furthermore, to better guide the masked region in the diffusion model, we introduce LoRA adapters (Hu et al., 2021). These adapters aid in learning the subject harmonization task:

$$\mathcal{L} := \mathbb{E}_{\boldsymbol{\varepsilon} \sim \mathcal{N}(0,1), t \sim \mathcal{U}(0,1)}[\|\boldsymbol{\varepsilon} + \boldsymbol{x} - \boldsymbol{x}^* - \boldsymbol{\varepsilon}_\theta([\boldsymbol{x}_t, \boldsymbol{m}, \boldsymbol{x}], t, c(\boldsymbol{z})\|_F^2], \qquad (2)$$

where $\boldsymbol{x}^*$ represents the target harmonized image, and $\boldsymbol{x}$ is the input image. While we tweak the training objective, the generation process of the diffusion models remains unchanged. This allows us to still utilize the pre-trained stable diffusion model with the learned harmonize-prompt and LoRA parameters, and seamlessly integrate with other modules. See Sec A.10 in the appendix for details..

## 2.5 ReS DATASET

To evaluate the effectiveness of subject repositioning algorithms, we curated a benchmark dataset called ReS. This dataset includes 100 paired images, each with dimensions $4032 \times 3024$, where one image features a repositioned subject while the other elements remain constant. These images were collected from over 20 indoor and outdoor scenes, showcasing subjects from more than 50 categories. This diversity enables effective simulation of real-world open-vocabulary applications. Thus our dataset is diverse enough to evaluate our SEELE model.

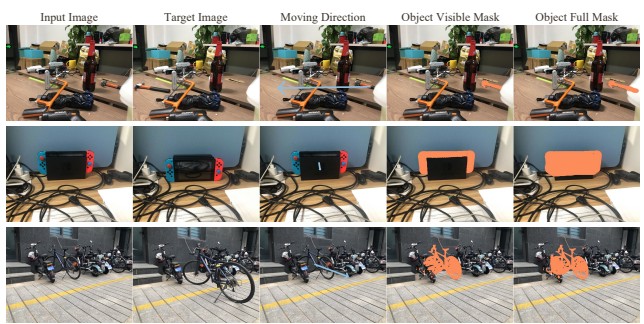

Figure 6: Examples of ReS dataset. The moving direction is marked as blue. The mask of visible part and completed subject specified by user are marked as orange.

The masks for the repositioned subjects were initially generated using SAM and refined by multiple experts. Occluded masks were also provided to assist in subject completion. The direction of repositioning was estimated by measuring the distance between the center points of the masks in each image pair.

For each paired image in the dataset, we can assess subject repositioning performance from one image to the other and in reverse, resulting in a total of 200 testing examples. Figure 6 illustrates the ReS dataset. We plan to release the ReS dataset to encourage research in subject repositioning.

## 3 RESULTS AND ANALYSIS

**Examples of subject repositioning**. We present subject repositioning results on $1024^2$ images using SEELE in Figure 7. SEELE works well on diverse scenarios of subject repositioning.

**Subject repositioning on ReS**. Since there are currently no publicly available models specifically designed for subject repositioning, we mainly compare with original Stable Diffusion inpainting model (SD). We adopt SD under no text prompts, simple prompts and complex prompts. The used prompts are provided in Sec. A.3 in the appendix. Furthermore, by combining masks from subject movement and completion sub-tasks into a single mask, we can incorporate alternative inpainting algorithms in SEELE. Specifically, we incorporate LaMa (Suvorov et al., 2021), MAT (Li et al., 2022), MAE-FAR (Cao et al., 2022), and ZITS++ (Cao et al., 2023) into SEELE. Note that in this experiment, SEELE does not utilize any pre-processing or post-processing techniques.

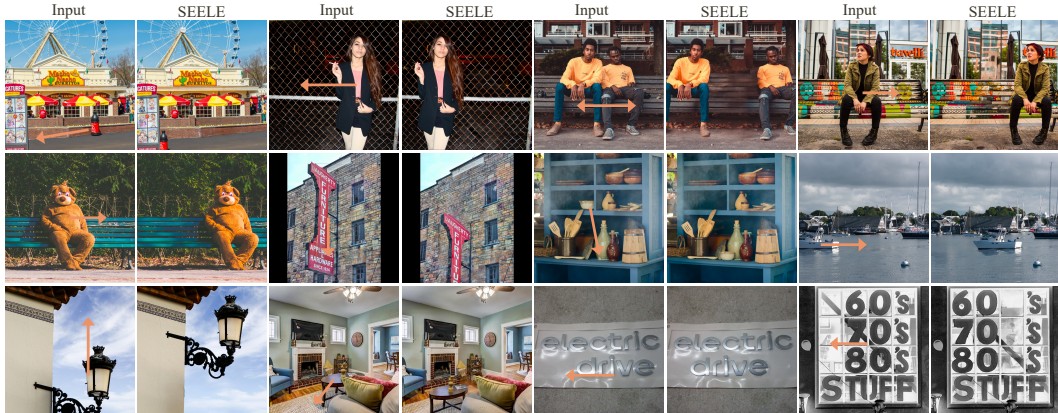

Figure 7: Subject repositioning on $1024^2$ images using SEELE. See larger version in Figure 13.

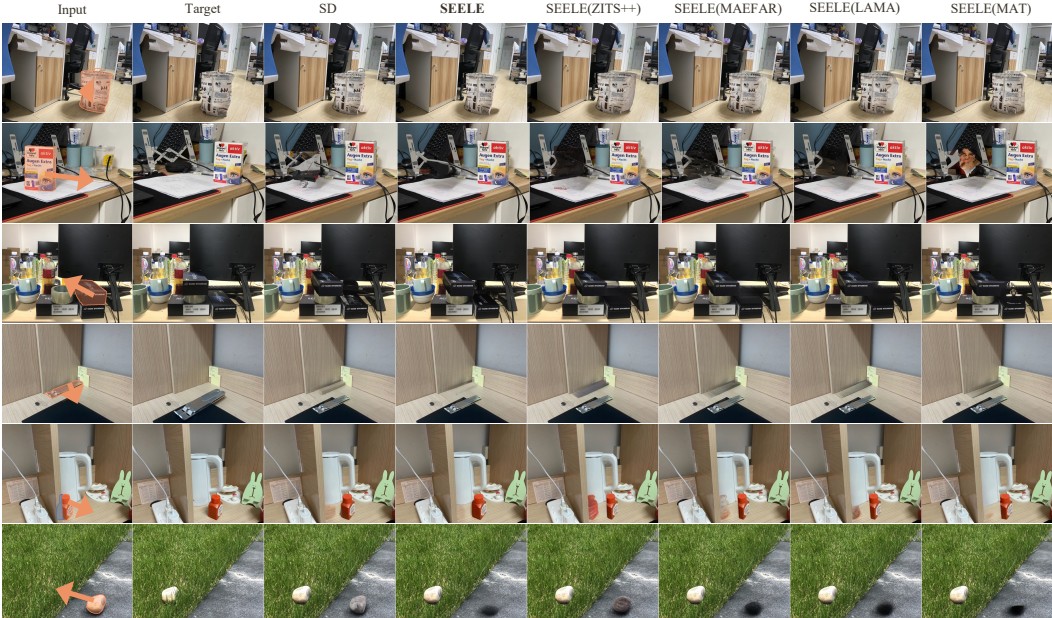

Figure 8: Qualitative comparison of subject repositioning on ReS.

We present qualitative comparison results in Figure 8 where a larger version is Figure 14 in the appendix. More results can be found in Figure 15 and Table 1 in the appendix. We add orange subject removal mask and blue subject completion mask in the input image. Our qualitative analysis indicates that SEELE exhibits superior subject removal capabilities without adding random parts and excells in subject completion. When the moved subject overlaps with the left void, SD fills the void guided the subject. In contrast, SEELE avoids the influence of the subject, as shown in the top row of Figure 8. If the mask isn't precise, SEELE works better than other methods by reducing the impact of unclear edges and smoothing out the empty space, as seen in the fourth row. Also, SEELE is excels in subject completion than typical inpainting algorithms, as seen in the second-to-last row. Note that SEELE can be further enhanced through the post-processing stage.

**Effectiveness of the proposed task-inversion**. To further validate the proposed task-inversion, we conduct experiments on standard inpainting and outpainting tasks, following the standard training and evaluation principles. We provide analysis in Sec. A.5 in the appendix where results for inpainting can be found at Table 2 and Figure 16 and outpainting at Table 3 and Figure 17.

**SEELE w/ X.** We assess the effectiveness of various components within SEELE during both pre-processing and post-processing phases. We conduct a qualitative comparison of SEELE's results with and without the utilization of these components, as shown in Figure 9 in the appendix, while a detailed analysis of each component is provided in Sec. A.4 in the appendix.

# 4    RELATED WORKS

**Image and video manipulation** aims to manipulate images and videos in accordance with user-specified guidance. Among these guidance, natural language guidance, as presented in previous studies (Dong et al., 2017; Nam et al., 2018; Li et al., 2020a;b; Xia et al., 2021; Karras et al., 2019; El-Nouby et al., 2019; Zhang et al., 2021; Fu et al., 2020; Chen et al., 2018; Wang et al., 2018; Jiang et al., 2021), stands out as particularly appealing due to its adaptability and user-friendliness. Some research efforts have also explored the use of visual conditions, which can be conceptualized as image-to-image translation tasks. These conditions encompass sketch-based (Yu et al., 2019; Jo & Park, 2019; Chen et al., 2020; Kim et al., 2020; Chen et al., 2021; Richardson et al., 2021; Zeng et al., 2022), label-based (Park et al., 2019; Zhu et al., 2020; Richardson et al., 2021; Lee et al., 2020), line-based (Li et al., 2019), and layout-based (Liu et al., 2019) conditions. In contrast to image manipulation, video manipulation (Kim et al., 2019; Xu et al., 2019; Fu et al., 2022) introduces the additional challenge of ensuring temporal consistency across different frames, necessitating the development of novel temporal architectures (Bar-Tal et al., 2022) . Image manipulation primarily revolves around modifying static images, whereas video manipulation deals with dynamic scenes in which multiple subjects are in motion. In contrast, our paper focuses exclusively on subject repositioning, where one subject is relocated while the rest of the image remains unchanged.

**Textual inversion** (Gal et al., 2022) is designed to personalize text-to-image diffusion models according to user-specified concepts. It achieves this by learning new concepts within the embedding space of text conditions while keeping all other parameters fixed. Null-text inversion (Mokady et al., 2022) learns distinct embeddings at different noise levels to enhance model capacity. Additionally, some fine-tuning (Ruiz et al., 2022) or adaptation (Zhang & Agrawala, 2023; Mou et al., 2023) techniques inject visual conditions into text-to-image diffusion models. While these approaches concentrate on image patterns, SEELE focuses on the task instruction to guide diffusion models.

**Prompt tuning** (Lester et al., 2021; Liu et al., 2021b;a) entails training a model to learn specific tokens as additional inputs to transformer models, thereby enabling model adaptation to a specific domain without fine-tuning the model. This technique been widely used in vision-language models (Radford et al., 2021; Yao et al., 2021; Ge et al., 2022). This concept has inspired us to transform the text-to-image diffusion model into a task-to-image diffusion model by tuning the text conditions.

**Image composition** (Niu et al., 2021) is the process of combining a foreground and background to create a high-quality image. Due to differences in the characteristics of foreground and background elements, inconsistencies can arise in terms of appearance, geometry, or semantics. Appearance inconsistencies encompass unnatural boundaries and lighting disparities. Segmentation (Kirillov et al., 2023), matting (Xu et al., 2017), and blending (Zhang et al., 2020) algorithms can be employed to address boundary concerns, while image harmonization (Tsai et al., 2017) techniques can mitigate lighting discrepancies. Geometry inconsistencies include occlusion and disproportionate scaling, necessitating object completion (Zhan et al., 2020) and object placement (Tripathi et al., 2019) methods, respectively. Semantic inconsistencies pertain to unnatural interactions between subjects and backgrounds and are beyond the scope of this paper. While each aspect of image composition has its specific focus, the overarching goal is to produce a high-fidelity image. In our paper, SEELE concentrates on enhancing harmonization capabilities within a single generative model.

# 5    CONCLUSION

In this paper, we introduce an innovative task known as subject repositioning, which involves manipulating an input image to reposition one of its subjects to a desired location while preserving the image's fidelity. To tackle subject repositioning, we present SEELE, a framework that leverages a single diffusion model to address the generative sub-tasks through our proposed task inversion technique. This includes tasks such as subject removal, subject completion, subject harmonization, and shadow generation. For the non-generative sub-tasks, we utilize pre-trained models. To evaluate the effectiveness of subject repositioning, we have curated a real-world dataset called ReS. Our experiments on ReS demonstrate the proficiency of SEELE in accomplishing this task.

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

# A APPENDIX

## A.1 ADDITIONAL EXAMPLES

In this section, we first present subject repositioning results on images of size $1024 \times 1024$ using SEELE in Figure 13. Then we provide a larger visualization of Figure 8 in our paper in Figure 14. Furthermore, we present additional examples of subject repositioning using SEELE and its competitors, as showcased in the proposed ReS dataset, within Figure 15.

## A.2 EXPERIMENTAL SETTING

SEELE is built upon the text-guided inpainting model fine-tuned from SD 2.0-base, employing the task inversion technique to learn each task prompt with 50 learnable tokens, initialized with text descriptions from the task instructions. For each task, we utilize the AdamW optimizer (Loshchilov & Hutter, 2017) with a learning rate of $8.0e - 5$, weight decay of $0.01$, and a batch size of 32. Training is conducted on two A6000 GPUs over 9,000 steps, selecting the best checkpoints based on the held-out validation set.

When addressing subject moving and completion, we employ the MSCOCO dataset (Lin et al., 2014), which provides object masks. For image harmonization, the iHarmony4 dataset (Cong et al., 2020) is utilized, offering unharmonized-harmonized image pairs along with subject-to-harmonize masks. MSCOCO comprises 80k training images and 40k testing images, while iHarmony4 includes 65k training images and 7k testing images. This diversity ensures robustness in training task prompts, guarding against overfitting on specific images.

**Cost analysis**. The core component of SEELE is the pre-trained stable diffusion inpainting model, boasting 865.93 million parameters within its UNet backbone. To tailor this stable diffusion model for subject repositioning, we incorporate three distinct task prompts, each sized at $50 \times 1024$ and has 0.5 million trainable parameters. For the local harmonization task, we introduce the LoRA adapter, which encompasses 5.12 million trainable parameters. It's worth noting that these newly added parameters are lightweight and introduce no additional inference latency when compared to the stable diffusion backbone.

## A.3 QUANTITATIVE COMPARISON IN ReS

To assess the effectiveness of subject repositioning, we conducted experiments on our newly introduced ReS dataset, employing various metrics, including Peak Signal-to-Noise Ratio (PSNR), Structural Similarity Index Measure (SSIM), and Learned Perceptual Image Patch Similarity (LPIPS). We resize the image to a shape with a minimum side length of 512 to fit standard inpainting algorithms. Regrettably, the Magic Editor feature of Google Photos is not available to the public at this time, and we do not have access to it. Therefore, we cannot compare SEELE with Magic Editor on ReS.

In comparison to the baseline stable diffusion model, SEELE demonstrates significant enhancements in the quality of manipulated images across all metrics, with particular emphasis on the PSNR. Employing a stable diffusion backbone with SEELE leads to improved LPIPS scores, while SEELE combined with standard inpainting algorithms yields superior PSNR results.

**SD prompts**. For SD with no prompt, we input with "" for the text condition. The simple prompts used for SD is "inpaint" and "complete the subject", as well as complex prompts such as "Incorporate visually cohesive and high-fidelity background and texture into the provided image through inpainting" and "Complete the subject by filling in the missing region with visually cohesive and high-fidelity background and texture" for subject movement and completion tasks, respectively.

## A.4 ANALYSIS OF X IN SEELE

Here we provide the analysis of each component used in SEELE.

i) *Depth estimation for occlusion* becomes crucial when users wish to move a subject from the foreground to the background. It helps estimate and correct the occluded parts, ensuring that the repositioned subject blends seamlessly into the scene. As illustrated in the first row of Figure 9, this depth estimation plays a pivotal role in repositioning objects like the tower behind leaves or people

Table 1: Quantitative comparison of subject repositioning.

| Model | PSNR(↑) | SSIM(↑) | LPIPS(↓) |
|---|---|---|---|
| SD (no prompt) | 20.038 | 0.664 | 0.157 |
| SD (simple prompt) | 20.039 | 0.664 | 0.157 |
| SD (complex prompt) | 20.029 | 0.664 | 0.157 |
| SEELE | 20.100 | 0.666 | 0.156 |
| SEELE(ZITS++) | 20.330 | 0.679 | 0.176 |
| SEELE(MAE-FAR) | 20.453 | 0.680 | 0.172 |
| SEELE(LaMa) | 20.320 | 0.678 | 0.163 |
| SEELE(MAT) | 20.199 | 0.675 | 0.163 |

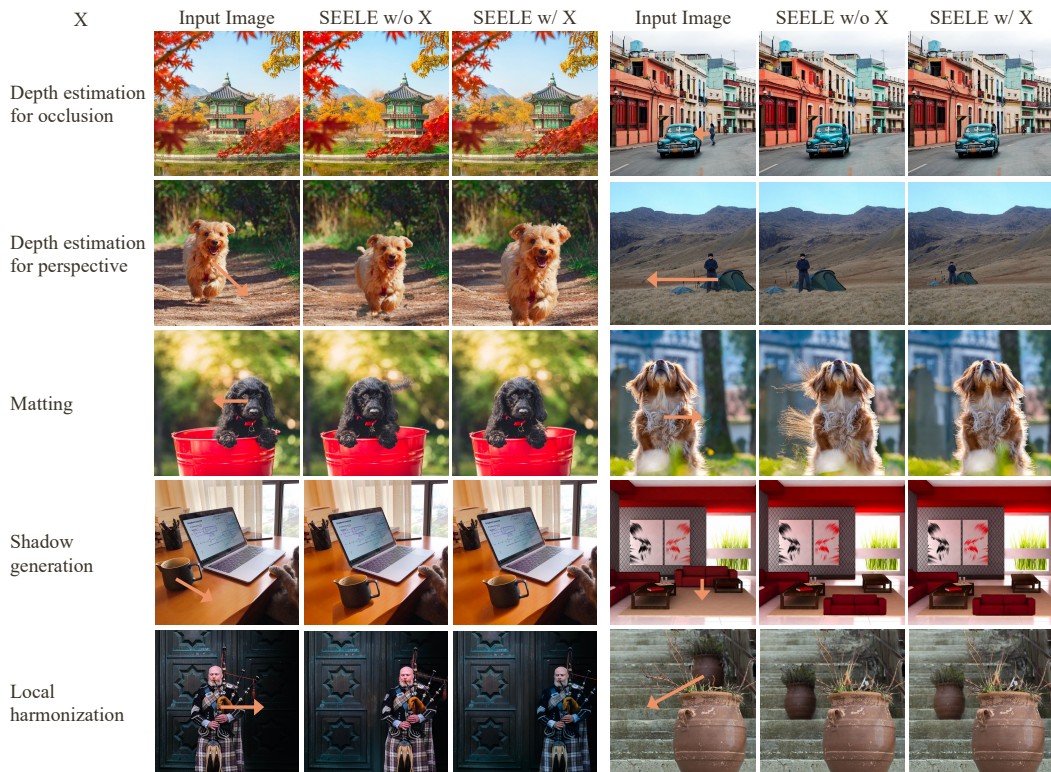

Figure 9: Ablation of using components X in SEELE.

behind a car. Neglecting the occlusion relationship can result in unnatural-looking repositioned subjects and a significant loss of image fidelity.

ii) *Depth estimation for perspective* comes into play when users want to resize the subject proportionally during repositioning. If this aspect is overlooked, the subject's size remains fixed, which may contradict user expectations.

iii) *Matting* primarily addresses issues arising from imprecise masks provided by SAM, particularly when dealing with subjects with ambiguous boundaries. Precise masking is crucial because inaccuracies can lead to information leaking in the final output. For example, in Figure 9, imprecise masking might encourage the gaps to generate unnatural dog fur.

iv) *Shadow generation* is handled by reusing the generative model within SEELE. In cases where a subject includes shadows, such as the left part in Figure 9, we approach it as a subject completion task. The shadow itself becomes the subject, and we employ a learned complete-prompt to guide

the diffusion model. Conversely, when a subject lacks shadows, we can transform it into a local harmonization task by utilizing SEELE's harmonization model to generate shadows.

v) *Local harmonization* addresses the challenge of appearance inconsistency. When the illumination statistics change after subject repositioning, it's essential to adjust the subject's appearance while preserving its texture. As depicted in Figure 9, SEELE excels at this local harmonization task, ensuring seamless integration into the new environment.

### A.5 STANDARD IMAGE INPAINTING AND OUTPAINTING

**Image inpainting**. The proposed task-inversion approach not only specializes the inpainting model for specific tasks but also enhances its standard inpainting capabilities. We substantiate this claim through experiments conducted on the Places2 dataset (Zhou et al., 2017), where we train SEELE using standard inpainting prompts and compare its performance with other inpainting algorithms. The results are presented in Table 2. Additionally, we provide visual representations of the results in Figure 16, demonstrating SEELE's advantage in reducing hallucinatory artifacts.

Table 2: Quantitative results for inpainting on Places2 (Zhou et al., 2017)

| Methods | PSNR↑ | SSIM↑ | FID↓ | LPIPS↓ |
|---|---|---|---|---|
| Co-Mod (Zhao et al., 2021) | 21.091 | 0.843 | 30.041 | 0.1664 |
| MAT (Li et al., 2022) | 20.680 | 0.838 | 32.439 | 0.1650 |
| SD (no prompt) | 20.353 | 0.839 | 29.632 | 0.1603 |
| SD ("background") | 20.591 | 0.844 | 29.313 | 0.1564 |
| SEELE | **21.982** | **0.869** | **24.401** | **0.1295** |

**Image outpainting**. Another commonly used manipulation task involves extending the image beyond its original content. This approach shares a similar concept with subject completion, but it takes a more holistic perspective by enhancing the entire image. We have also conducted experiments on the outpainting task and demonstrated the effectiveness of task inversion. Our experiments were carried out using the Flickr-Scenery dataset (Cheng et al., 2022), and the results are compared with stable diffusion in Table 3. The results indicate the superiority of task inversion employed in SEELE. Furthermore, we provide visual examples for qualitative assessment in Figure 17.

Table 3: Quantitative results for outpainting on Flickr-Scenery (Cheng et al., 2022).

| Methods | PSNR↑ | SSIM↑ | FID↓ | LPIPS↓ |
|---|---|---|---|---|
| SD (no prompt) | 14.476 | 0.693 | 53.523 | 0.3475 |
| SD ("background") | 14.601 | 0.696 | 46.582 | 0.3429 |
| SEELE | **15.989** | **0.731** | **29.056** | **0.3131** |

### A.6 ABLATION OF USING THE OPPOSITE TASK PROMPTS FOR DIFFERENT SUB-TASKS

The introduced task-specific prompts guides the different generation directions. Hence if we use the opposite task prompts for different sub-tasks, an opposite direction will leads to the diffusion model to produce unreasonable results. To show this, we conduct an qualitative comparison of using different task prompts in Figure 10. Evidently, the opposite task prompts will leads to the failure of the corresponding sub-tasks.

In the first example, we use an imperfect segmentation mask to highlight the different behaviour of different prompts. The remove-prompt, trained to re-paint the hole uniformly guided by the surrounding areas, accurately generate flowers in the masked region. On the other hand, the complete-prompt, trained to complete the adjacent subject, identifies the mask as part of a fly, thus trying to generate the fly in the masked region and reduce the influence of the flowers.

In the second example, when we want to complete the bird head. The remove-prompt simply gener-

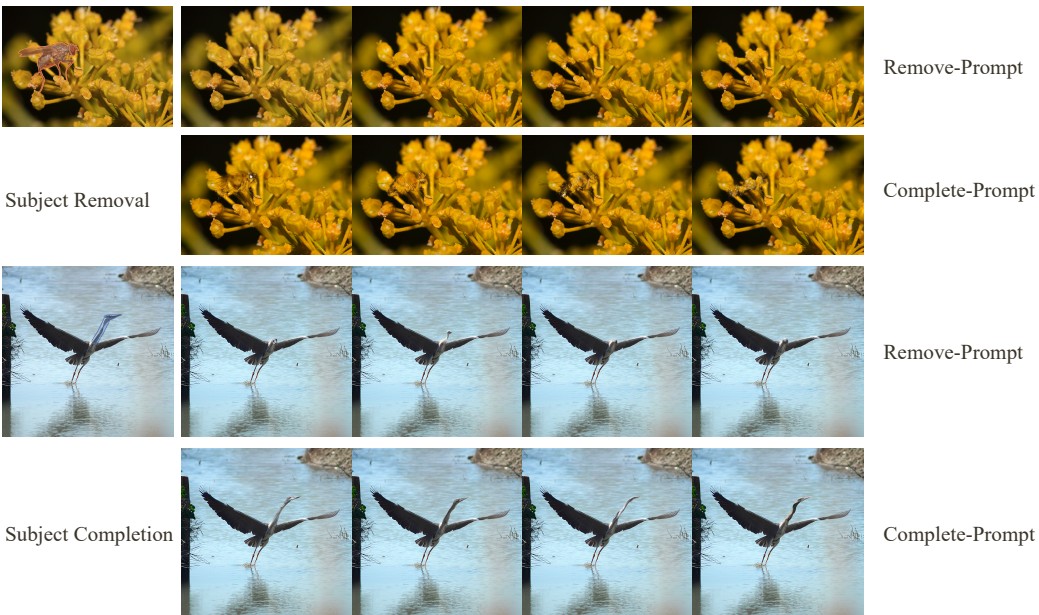

Figure 10: Ablation of using the opposite task prompts for different sub-tasks.

ate waters guided by the surroundings, while complete-prompt accurately captures the semantics of the masks as part of the bird and generate the bird head.

### A.7  ABLATION OF LOCAL HARMONIZATION

To tackle the local harmonization sub-task, we learn the harmony-prompt along with the LoRA parameters. To show the efficacy of each module, we conduct an qualitative ablation study in Figure 11. Naturally, if we disable the LoRA parameters, as we use the inharmonious image as unmasked image condition for the stable diffusion model, the model tends to copy the image without significant modification. If we only use LoRA parameter, it works like the unconditional diffusion model to perform local harmonization, but usually performs over- or under- harmonization. Such a manner works to some extent, but can be enhanced with the learned harmony-prompt.

### A.8  VISUALIZATION OF SAM EVERYTHING MODE

As we use the everything mode in SAM when user input text to select the subject, here we provide visualization of segmentation results of SAM everything mode for convenience. To extract subject-level mask from SAM everything mode, we filter the generated masks of SAM with a preference of preserving larger mask, then compare the feature of masked region with CLIP text model to identify the most similar part. As SAM cannot produce object-level masks directly, in practice, an iterative refinement following user guidance is usually required.

### A.9  NECESSARITY OF USING DIFFERENT DATASETS TO TRAIN SEELE

Our training of the SEELE model utilized only two datasets: COCO, which provides ground-truth object segmentation masks, and iHarmony4, which offers paired images for local harmonization tasks. These datasets, chosen for their public availability, aptly fulfill the varying requirements of different generative sub-tasks. Our training approach, which encompasses both subject movement and completion, employs a unified task inversion technique. Given that local harmonization focuses

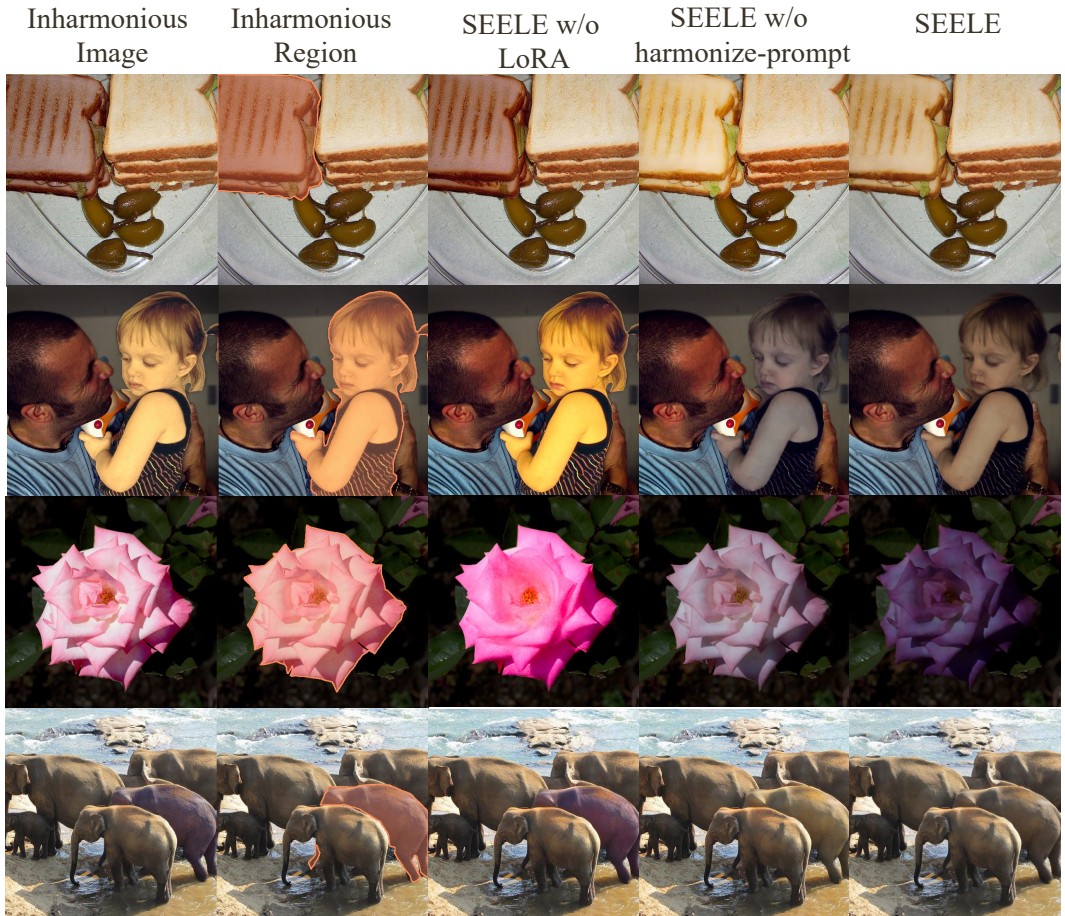

Figure 11: Ablation of local harmonization.

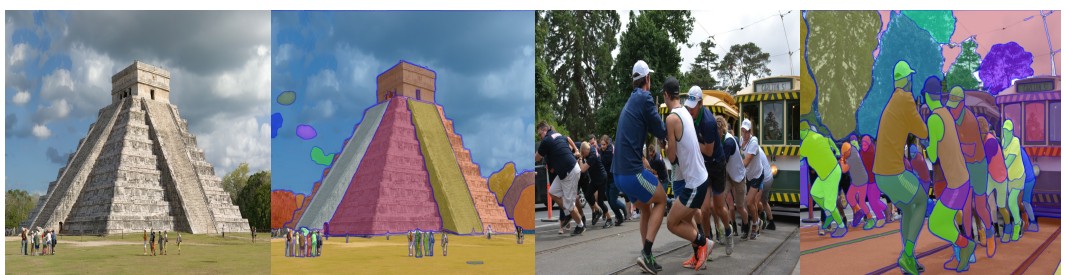

Figure 12: Visualization of SAM everything mode.

on not introducing new details in masked areas, we have modified the diffusion model to integrate the characteristics of the masked region, ensuring it aligns with the task's specific needs.

## A.10 INTEGRATING LoRA

When the LoRA adapter is trained, we load them along with the frozen stable diffusion model. As LoRA is implemented as additive layers with the original layers. For example, suppose for a particular layer $f$ with input $x_i$ and output $x_{i+1}$. The original stable diffusion performs $x_{i+1} = f(x_i)$, while LoRA is trained to perform $x_{i+1} = f(x_i) + \text{LoRA}(x_i)$ and only learn $\text{LoRA}(\cdot)$ while freezing $f(\cdot)$. Then we could introduce a scale hyper-parameter for a trained model $x_{i+1} = f(x_i) + c\text{LoRA}(x_i)$ When SEELE performs the sub-tasks in manipulation process, we set the lora scale as $c = 0$ to preserve the original outputs of stable diffusion. While in the local harmonization process, we set the lora scale as $c = 1$ to perform local harmonization. In this regard, we could use the same stable diffusion backbone and perform different sub-tasks using different sub-task prompts (and LoRA parameters).

## A.11 LIMITATIONS

One significant limitation of SEELE is that when the system performs suboptimally, manual user intervention becomes necessary to enhance the results. For instance, in cases where segmentation fails, users are required to manually correct the segment mask. Similarly, when the subject is occluded, users must provide a mask of potential regions to complete the subject. The former issue could potentially be mitigated through improvements in the segmentation model. However, the latter challenge necessitates the development of a novel model to address the problem of open-vocabulary amodal mask generation (Zhan et al., 2020). Currently, there is a lack of available foundation models to support open-vocabulary amodal mask generation, as well as a scarcity of large datasets suitable for training such a model. These aspects are considered potential avenues for future research.

## A.12 WEB USER INTERFACE (WEB-UI)

In this section, we provide an overview of SEELE's front-end user interface (UI), which users interact with when utilizing SEELE. This web-based UI has been designed based on Gradio (Abid et al., 2019) and is depicted in Figure 18.

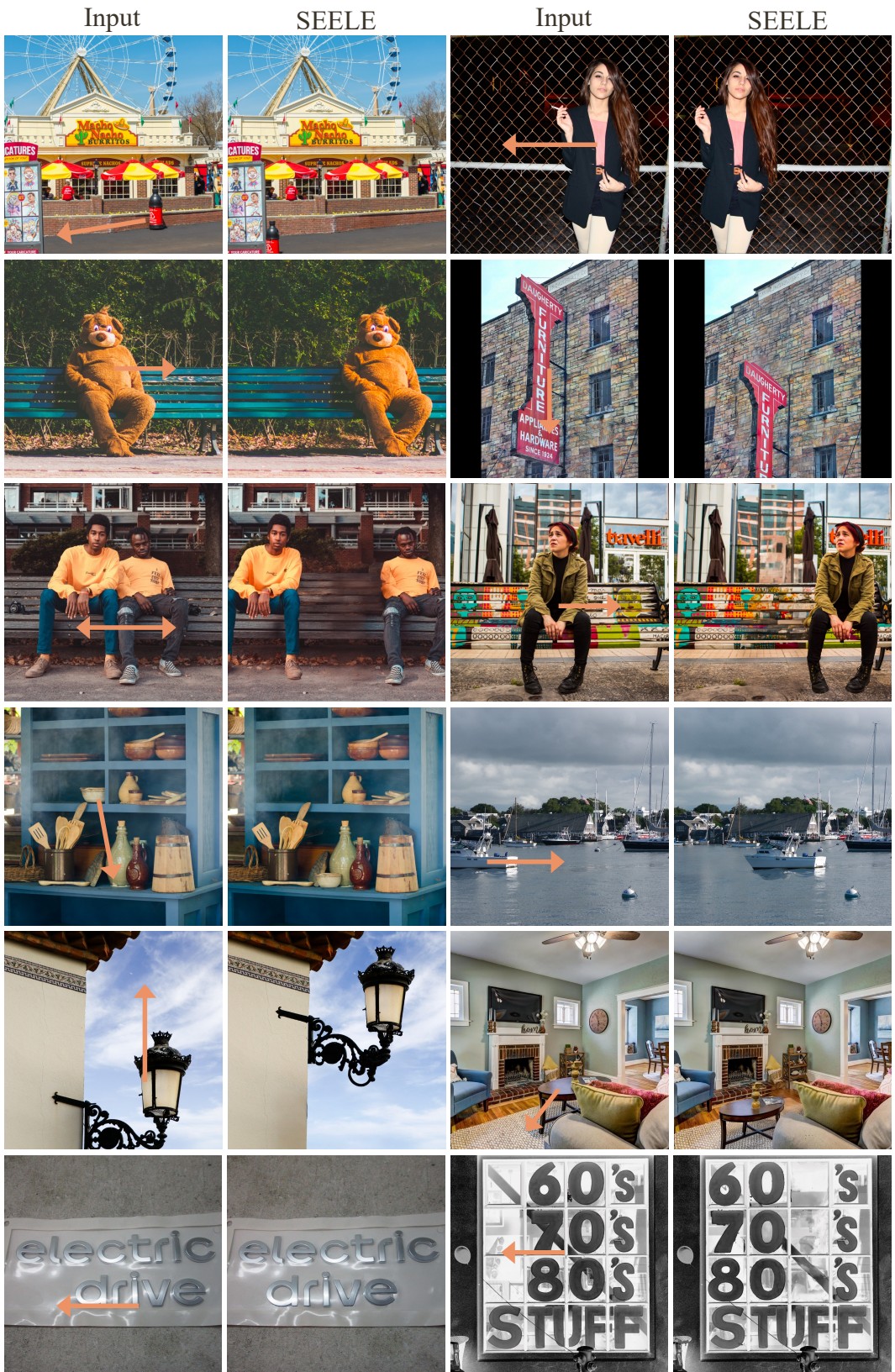

Figure 13: SEELE on images of size $1024 \times 1024$.

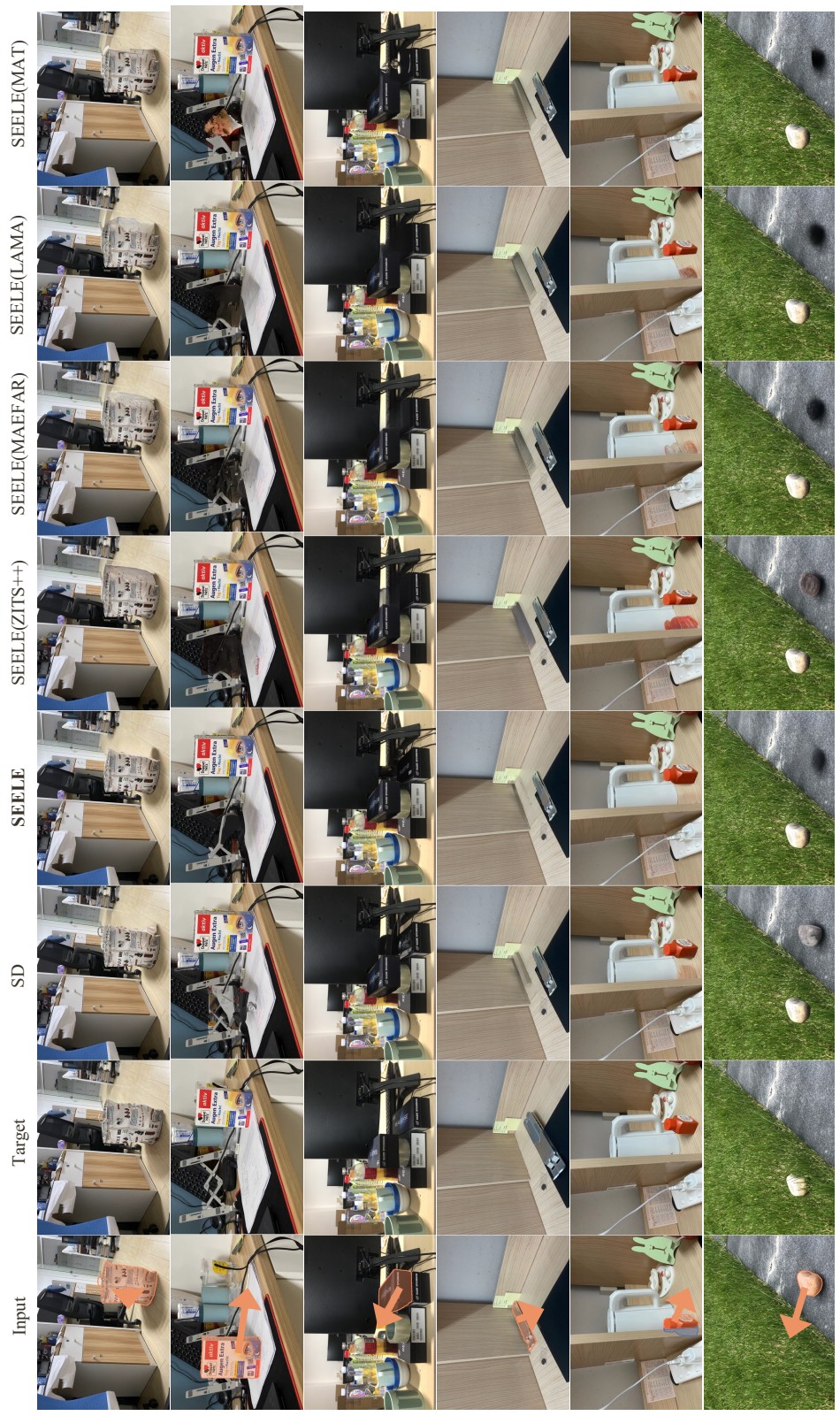

Figure 14: Qualitative comparison for subject repositioning in ReS.

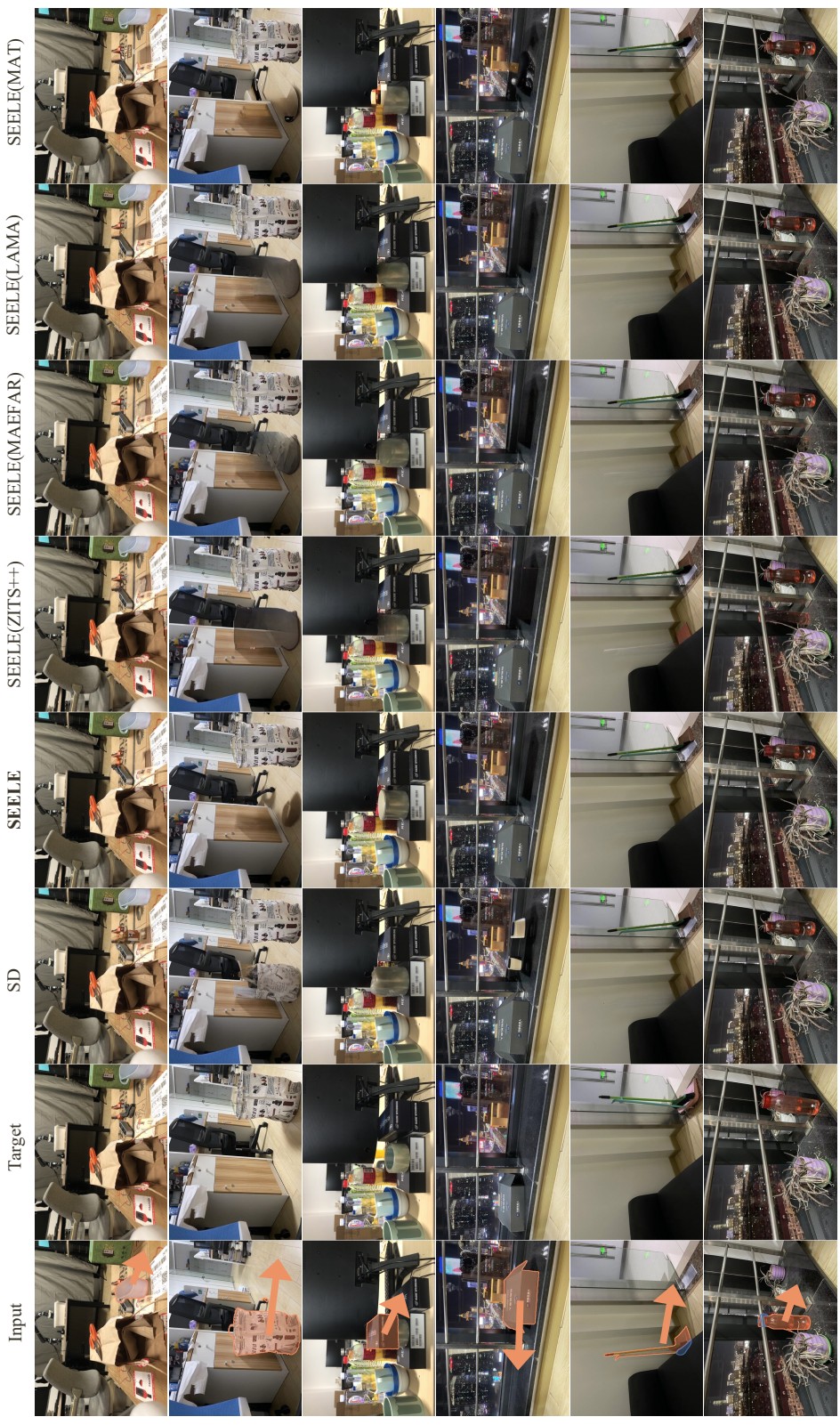

Figure 15: More qualitative comparison for subject repositioning in ReS.

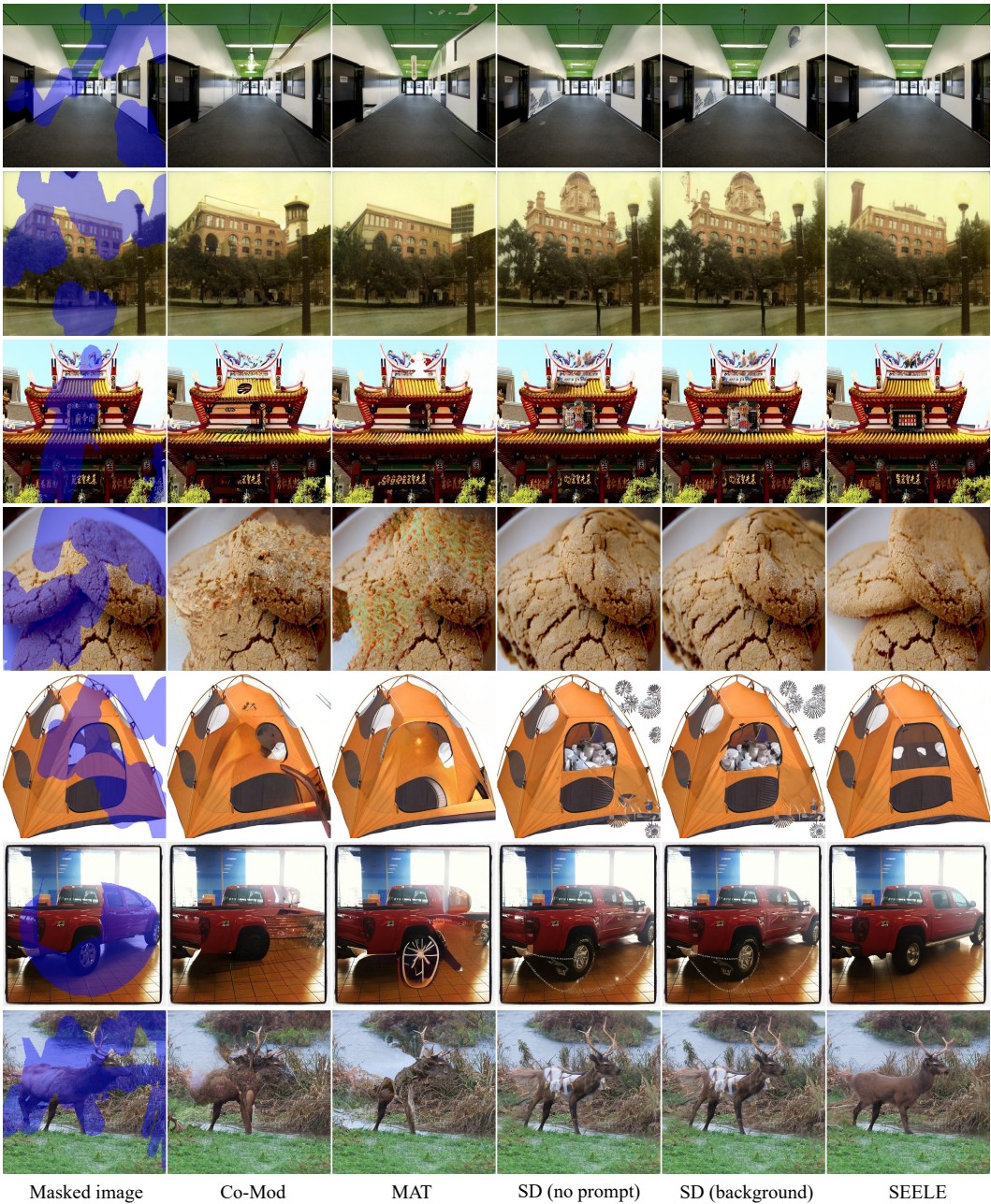

Masked image     Co-Mod     MAT     SD (no prompt)     SD (background)     SEELE

Figure 16: Qualitative comparison for inpainting.

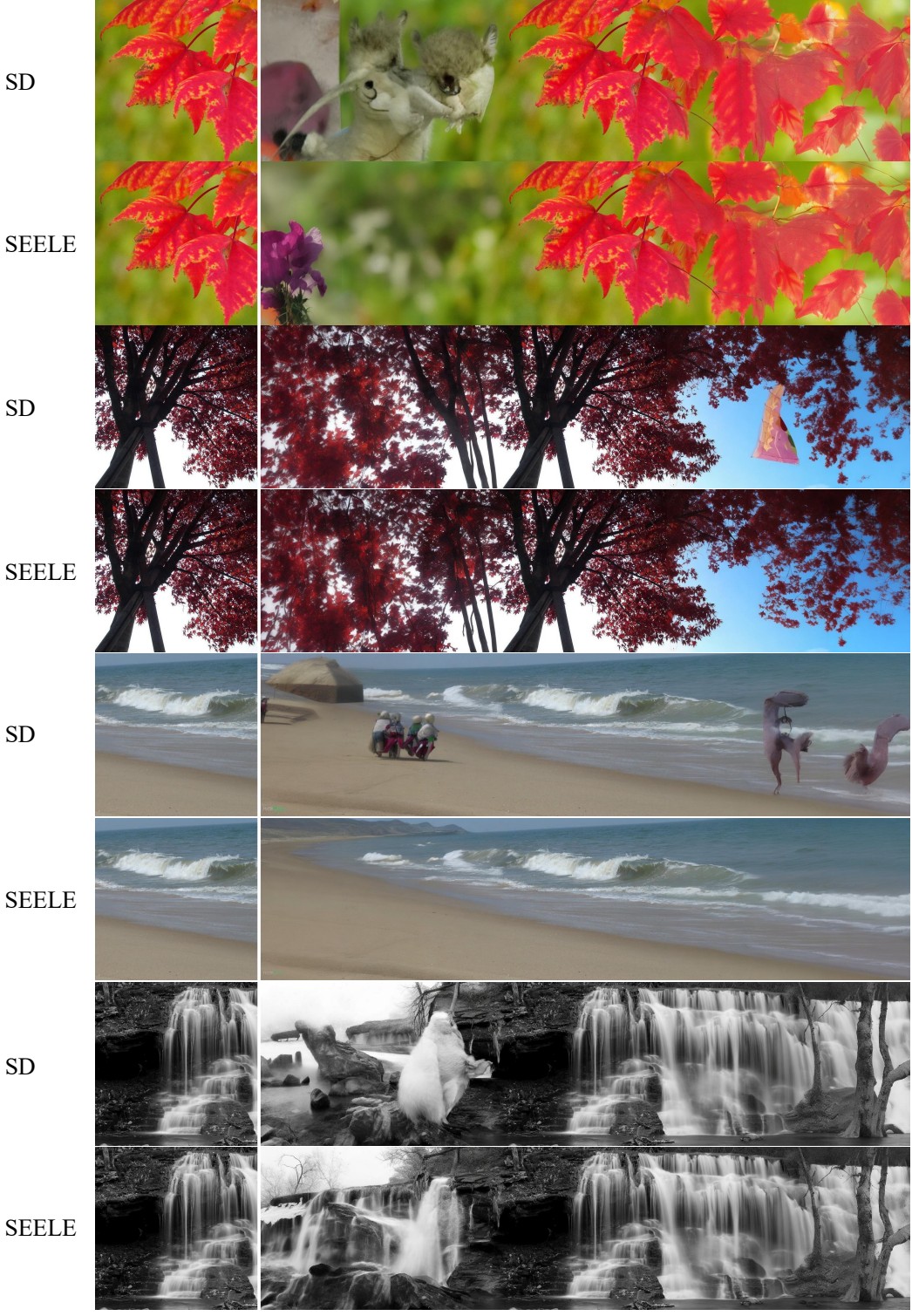

Figure 17: Qualitative comparison for outpainting.

Figure 18: Web-UI for SEELE.

