# OpenReview forum: "Repositioning the Subject within Image"
_ICLR.cc/2024/Conference — ICLR 2024 Conference Withdrawn Submission_

### Official Review · Reviewer_RbUq · 2023-10-25

**Soundness:** 3 good
**Presentation:** 2 fair
**Contribution:** 2 fair
**Rating:** 3
**Confidence:** 5

**Summary:**

This paper proposes to reformulate the task of repositioning the subject within an image as a unified prompt-guided inpainting task. Specifically, different sub-tasks are achieved by the proposed task inversion technique, as well as pre/post-processing techniques. To verify the effectiveness, the authors also propose a real-world subject repositioning dataset.

**Strengths:**

1. The authors provide a detailed analysis and a formulation for the target of repositioning the subject within an image, the proposal is interesting.

2. This paper shows some promising results in terms of repositioning the subject.

3. The authors collect 100 real image pairs for object repositioning. Specifically, the masks are claimed refined by experts. This dataset can be useful to the community for further applications.

**Weaknesses:**

1. The presentation of this paper needs grand improvements. For example,

    1. The organization of this paper is confusing. The authors formulate the repositioning task as three sub-tasks, i.e., local inpainting, subject completion, and local harmonization, and try to solve them by task inversion. However, They only introduce subject moving and completion in Section 2.3 and consider harmonization as post-processing in Section 2.4. Besides, the authors introduce the task in Section 2.3 but leave the whole picture until Section 2.4 which introduces both SEELE framework and harmonization. The organization can make readers hard to follow.

    2. I think Figure 2, the pipeline overview is important for readers to catch up with the whole picture. However, there are multiple input and output lines in the Manipulation Model, which also confuses me a lot, what input and output for which sub-task? Besides, in Figure 2, harmonization is in the post-processing step beyond the scope of the manipulation model, which conflicts with the claim as ‘a unified single model‘ in the Introduction.

    3. Many important details are missing in the paper. For every qualitative result, the authors should denote the moving mask, and completion mask in the inputs.


2\. The motivation/novelty is not that convincing to me. Here are some justifications,

1. The authors claim that they “introduce an innovative dynamic manipulation task”, however, Zhan et al. [1] have introduced the same task and a complete pipeline (also contains local inpainting and subject completion) for the same target. What’s more, I didn’t see much superiority of SEELE over Zhan’s solution given that the proposed SEELE requires many manual efforts. The authors should discuss and compare their solutions.

2. local inpainting and subject completion are actually can be unified as completing the region according to context (background or object). Why do we need to explicitly split them into two distinct sub-tasks? What’s the effect if we use opposite task prompts for different sub-tasks?

3. The optimization objectives in EQ(2) is confusing. Why do we need an extra term, i.e., x - x\\*?

4. In section 2.3, the authors claim that they learn move-prompt for subject-moving tasks. however, the optimization target is more like local inpainting instead of the subject moving, which also involves moving and blending the subject in a new place.


3\. I’m afraid the experimental results are not strong enough to support the paper’s claim. Specifically,

1. Since the authors use both task inversion and LoRA adapters for local harmonization, we suggest the authors conduct an ablation study to verify the necessity of using LoRA and the effectiveness of task inversion here.

2. The inpainting results of SEELE seem not strong enough. For example, the wood texture looks blurry and flattened in Fig. 1. The perspective issue remains in the SEELE in the 4th case in Figure 6 and most results in Figure 6 look similar. Besides, the results of LAMA and MAT in the 5th case in Figure 6 look weird. It seems that LAMA and MAT didn’t perform subject completion as they have very straight boundaries of the subject.

3. The authors introduce that they use SAM to segment the entire image into distinct subjects in the pre-processing step. We suggest the authors show the SAM results for readers as a reference.

4. The comparison in Figure 11 is not fair. Specifically, the authors should also show the results of SD by using object prompts given that SEELE already requires many user efforts. Besides, it is not clear how to get the results of SEELE in Figure 11, by local inpainting or subject completion.

**Questions:**

See more details in the Weaknesses. I'm looking forward to the authors' response that addresses my concerns listed in the Weaknesses and am willing to raise my rating upon the response.

---

> ### Author Response · Authors · 2023-11-11
> **Response to Reviewer RbUq (1/3)**
>
> Dear Reviewer RbUq,
> Your thoughtful and detailed comments have provided us with an opportunity to reflect deeply on our work and enhance its quality. We are grateful for your guidance and will diligently incorporate your feedback into our revision. If our responses have addressed your concerns, please consider revising your rating accordingly.
>
> **1. Organization of methodology part is confusing.**
>
> Thanks for your valuable comment.
>
> 1) We reserve local harmonization for post-processing, as its purpose is not to generate new elements (either in the background or the subject), but rather to ensure consistency between masked and unmasked regions.
>
> 2) We appreciate your suggestion and we plan to revise our approach by presenting an overview of the entire concept before delving into the detailed components in our upcoming revision.
>
>
> **2. Figure 2 needs clarification.**
>
> We apologize for any confusion caused. During the manipulation phase, we demarcate the inputs and outputs for each specific sub-task using dashed boxes. The image is central to our process; at each step, we input an image and the resulting output image is depicted by directional arrows. In the 'subject moving' and 'subject completion' sub-tasks, we further illustrate the 'move prompt' and 'complete prompt' to emphasize the uniform application of our manipulation model across all generative sub-tasks.
>
> To address any misunderstanding regarding the consistency of our manipulation model in the post-processing stage, we confirm that the same model is indeed utilized throughout. We will make necessary amendments to Figure 2 in our revised document to better convey this.
>
> **3. Masks needs to be provided in qualitative result.**
>
> Thanks for the suggestion, we will follow your suggestion to add the masks in our qualitative results.

---

> ### Author Response · Authors · 2023-11-11
> **Response to Reviewer RbUq (2.1/3)**
>
> **1. Difference between SEELE and Zhan et al. [1].**
>
> We would like to emphasize the difference between Zhan et al.[1] and our proposed algorithm. In the following, we will refer to the algorithm in Zhan et al. [1] as scene de-occlusion.
>
> 1) Input Requirements: Scene de-occlusion necessitates all object masks in the image to establish object order relationships, whereas SEELE only requires users to indicate the subject of interest. This distinction is critical for two reasons:
>    i) Presently, no robust, open-vocabulary model exists for automated object-level segmentation to provide comprehensive object masks;
>    ii) Demanding users to label an entire image for subject repositioning is both impractical and unnecessary.
> Therefore, despite scene de-occlusion addressing similar tasks as subject repositioning, it is not feasible under user-friendly conditions like those offered by SEELE.
>
> 2) Task Focus: Scene de-occlusion is centered on understanding scenes and recovering the order of objects, which allows for object movement and amodal completion. This aspect aligns closely with SEELE, hence the citation for this specific sub-task. However, scene de-occlusion does not cater to subject movement tasks like local harmonization or shadow generation, nor does it handle subjects comprising multiple objects. In contrast, SEELE methodically addresses the complete process of subject repositioning.
>
> 3) Pipeline Differences: The workflows of scene de-occlusion and SEELE are fundamentally distinct. Scene de-occlusion prioritizes reconstructing object order and completing amodal masks for occluded objects, followed by generating these amodal objects. Conversely, SEELE exclusively concentrates on the user-specified subject, treating the rest of the image as background. It involves moving the subject, filling voids, completing the subject, and, if needed, performing shadow generation and local harmonization. It's not accurate to claim that both systems have a complete pipeline for the same objective, as they address different stages, targets, and methodologies.
>
> Comparison Limitations: While we strive to compare SEELE with scene de-occlusion, our dataset's lack of object-level masks poses a challenge for this analysis during the rebuttal period. It's important to note that SAM segmentation masks are not controlled to object-level, making the acquisition of such masks for all objects in an image a complex task.
>
> **2. Why consider local inpainting and subject completion as two distinct sub-tasks?**
>
> Thank you for your feedback. We concur that these sub-tasks can essentially be considered as variations of the same task. Approaches like the original Stable Diffusion or other inpainting algorithms demonstrate competent performance, as evidenced by our qualitative and quantitative analyses. However, a more detailed examination reveals distinct requirements for each sub-task.
>
> Specifically:
>
> 1) For region completion in line with the background, masks typically adopt object-like shapes. This often results in the generation of new, random subjects, influenced by the shape of the mask. This phenomenon is commonly observed in the Stable Diffusion model. As SEELE is built upon Stable Diffusion, we employ a task inversion technique to mitigate this tendency.
>
> 2) Although completing backgrounds and objects may seem similar, they necessitate different approaches to information use. Background completion, typically non-semantic, should uniformly influence the masked area. Conversely, objects are rich in semantics, and the model is expected to integrate the majority of the masked region as part of the object.
>
> When employing reverse task prompts, we might notice arbitrary semantic generation in the moving subject sub-task and suboptimal subject filling in the object completion sub-task. We will include a comparison in our revised submission to illustrate these points.

---

> ### Author Response · Authors · 2023-11-11
> **Response to Reviewer RbUq (2.2/3)**
>
> **3. Clarification on Eq(2).**
>
> Our goal in local harmonization is to ensure consistency between masked and unmasked regions.
> We utilize an inharmonious image as the input, denoted as \( x \).
> However, adhering to the standard training objective of stable diffusion results in a similarly inharmonious target, which is contrary to our aim of achieving a harmonious image.
> Therefore, we refocus the training objective of the diffusion model to directly produce a harmonious image from an inharmonious one, necessitating the addition of an extra term.
>
> **4. Clarification on learn move-prompt for subject-moving tasks**
>
> We apologize for any confusion caused.
> The process of subject-moving within SEELE involves duplicating the subject to the chosen location and then concealing the original position. This action creates a void that needs to be filled with the background. To address this, we have developed a specialized task prompt specifically designed for background integration. This task arises due to the displacement of the subject, hence we refer to it as the subject-moving task. We intend to provide a more detailed explanation in our revised document. In terms of seamlessly integrating the subject into a new location, we divide this task within SEELE into two phases: the subject completion sub-task and the subsequent post-processing stage. Our proposed methods effectively address these steps, and we will elaborate further in our upcoming revision.

---

> ### Author Response · Authors · 2023-11-11
> **Response to Reviewer RbUq (3/3)**
>
> **1. Ablation study to verify the necessity of using LoRA and the effectiveness of task inversion.**
>
> Thanks for the suggestion, we will add this ablation study in our revision.
>
> **2.The inpainting results of SEELE seem not strong enough.**
>
> Thank you for your feedback.
>
> 1) In Figure 1, SEELE demonstrates superior color consistency compared to Google Magic Editor, though it is not flawless. Our primary goal is to establish an innovative framework that either matches or exceeds the performance of Google Photos' latest AI feature for repositioning subjects within images. Our approach relies on a single stable diffusion model, employing diverse task prompts for varying generative directions, which enhances performance relative to the original stable diffusion, as verified in our experiments. While a more advanced generation model could potentially address specific sub-tasks more effectively, exploring such models falls outside the scope of this paper.
>
> 2) For an equitable comparison in Figure 6, we excluded the depth estimation algorithm from SEELE, thus not addressing perspective issues. Concerning the results from LAMA and MAT, the mask we used, created by SAM and refined by experts at high resolution, may leave some subject boundaries unmasked in lower resolutions. This problem could be mitigated by expanding the mask area, but in this instance, we did not focus on such specific issues for individual images, as all methods were applied to the same input image and mask.
>
> **3. Visualization SAM.**
>
> Thanks for the suggestion. We will add some visualization in our revision.
>
> **4. Qualification on Figure 11.**
>
> Apologies for any confusion. We would like to emphasize that the results shown in Figure 11 were obtained through standard inpainting, which was performed without requiring any user intervention. As outlined in Section A.5, SEELE adheres to the conventional inpainting pipeline for this task. It was trained using randomly generated masks to demonstrate the efficacy of our proposed task inversion technique.
> On the other hand, it is unfair to use image-dependent object prompts for SD, as SEELE do not involve any user efforts in inpainting task.
> Please note that Figure 11 is used to validate the effectiveness of our proposed task inversion technique.
> We will ensure to make this clear in our revised submission.

---

> ### Author Response · Authors · 2023-11-13
> **Ablation study submitted**
>
> Dear Reviewer RbUq,
>
> In response to your valuable suggestions, we have incorporated the proposed ablation studies into our revision. For your convenience, here is a brief outline:
>
> i) Problem 2.2 can be found in Section A.6 of the appendix.
>
> ii) Problem 3.1 is located in Section A.7 of the appendix.
>
> iii) Problem 3.3 is presented in Section A.8 of the appendix.
>
> Please inform us of any additional concerns.
>
> Sincerely,
>
> The Authors.

---

### Official Review · Reviewer_RPVi · 2023-10-27

**Soundness:** 3 good
**Presentation:** 2 fair
**Contribution:** 3 good
**Rating:** 5
**Confidence:** 4

**Summary:**

This paper proposes a segment-generate-and-blend framework with a preprocessing, manipulation, and post-processing pipeline to address the challenging object reposition task in a single image. In addition, the authors also assembled a real-world subject repositioning dataset called ReS which consists of 100 real image pairs featuring a repositioned subject. The main contribution of this work is to convert task instructions to a text embedding in the text-condition space. Besides, it employs other SOTA preprocessing (image segmentation and occlusion relationship estimation) and post-processing algorithms to optimize the whole object reposition performance.

**Strengths:**

1. This is the first framework after the deep learning era that deals with the object reposition issue in a still image.
2. This paper defines the object reposition task as a comprehensive image manipulation task that has several sub-tasks, including local inpainting, subject completion, and local harmonization, which is a new perspective.

**Weaknesses:**

1. There is an object reposition paper [1] before the deep learning era, which should be mentioned by you.
2. I think the most important part and contribution is the definition of task inversion in Sec. 2.2. But I still don't understand how to "translate" users' intentions into the so-called task embedding (task-specific prompts). How do you represent the "arrows" in figures 6 and 7 as task embeddings? The key step is not very clear in your paper.
3. You introduce a LoRA adapter when learning local hormonization, but how do you integrate it with the manipulation process? The statement is not very clear.


[1] Iizuka, Satoshi, et al. "Object repositioning based on the perspective in a single image" Computer Graphics Forum, Vol. 33, No. 8, 2014.

**Questions:**

1. In Figure 2, I don't identify the difference between the completed image and the output image after postprocessing. Can you provide a further explanation?
2. In Figure 4, how to obtain the mask for object completion as shown on the right side? The description is not very clear.
3. In the example shown in Figure 1 above, how can you determine whether to move the little girl only or the ballon and chair together?

---

> ### Author Response · Authors · 2023-11-11
> **Response to Reviewer RPVi**
>
> Dear Reviewer RPVi,
> Thank you for your rigorous review and insightful comments on our submission. We appreciate your contribution to enhancing the scientific rigor of our work. We have thoroughly addressed your suggestions and will revise accordingly. If our amendments align with your expectations, please consider re-evaluating the rating.
>
> Here are our responses to your insightful comments:
>
> **1. Missing citation**
>
> Very thanks for the suggestion. We will cite this paper and discuss it in our revision.
>
> **2. Clarification on users' intentions and task embedding.**
>
> Apologies for any confusion caused. In our system, the movement intentions of the users are represented as a vector comprising directional movements. This can be manually manipulated through inputs such as the "Move X" and "Move Y" options in our Gradio demo shown in Figure 13, or by dragging the chosen subject to a specified location.
>
> In this approach, the subject is moved in accordance with the defined direction and then replicated at the intended spot. It's important to note that this process is non-generative; the challenge lies in filling the resultant empty space. This is considered a generative sub-task, which we address through our unique subject-moving, prompt-guided generation technique. We intend to provide more clarity on this in our revised documentation.
>
>
> **3. How do you integrate LoRA with the manipulation process?**
>
> Sorry for the confusion.
> When the LoRA adapter is trained, we load them along with the frozen stable diffusion model.
> As LoRA is implemented as additive layers with the original layers. For example, suppose for a particular layer $f$ with input $x_i$ and output $x_{i+1}$.
> The original stable diffusion performs $x_{i+1}=f(x_i)$, while LoRA is trained to perform $x_{i+1}=f(x_i)+\textrm{LoRA}(x_i)$ and only learn $\textrm{LoRA}(\cdot)$ while freezing $f(\cdot)$.
> Then we could introduce a scale hyper-parameter for a trained model $x_{i+1}=f(x_i)+c\textrm{LoRA}(x_i)$
> When SEELE performs the sub-tasks in manipulation process, we set the lora scale as $c=0$ to preserve the original outputs of stable diffusion.
> While in the local harmonization process, we set the lora scale as $c=1$ to perform local harmonization.
> In this regard, we could use the same stable diffusion backbone and perform different sub-tasks using different sub-task prompts (and LoRA parameters).
>
> **4. Difference between the completed image and the output image after postprocessing in Figure 2.**
>
> Apologies for any confusion caused. Figure 2 serves to demonstrate the SEELE pipeline. Regarding the specific image in question, since the final output already exhibits a harmonious composition, we opted not to apply any postprocessing. We intend to substitute the current illustration with an example that includes a postprocessing step. The impact and effectiveness of the postprocessing phase, particularly in terms of shadow generation and local harmonization, are detailed in our ablation study presented in Figure 7.
>
> **5. Clarification on masking strategy for object completionin Figure 4**
>
> Thank you for your feedback. As detailed in Section 2.3, under the 'Subject Completion' section, our approach to mask construction is a two-step process:
> 1) Initially, we select a subject at random within the image and obtain its segmentation mask. A portion of this mask is then randomly chosen.
> 2) Subsequently, we apply random dilation to the mask, aiming to simulate the imprecise masking that might be generated by users.
> We will update Section 2.3 to make these steps clearer and more comprehensible.
>
> **6. In the example shown in Figure 1 above, how can you determine whether to move the little girl only or the ballon and chair together?**
>
> Thank you for your feedback.
> In our approach, we prioritize user intentions to determine the subject for repositioning.
> Specifically, we utilize the SAM as our segmentation model, allowing users to specify the subject through text, a point, or a bounding box.
> For instance, a user can indicate a subject by pointing to the little girl or by typing "little girl" initially.
> Should a user find the segmentation outcome unsatisfactory, such as SAM masking only the little girl and omitting the balloon and chair, they can refine the mask by pointing to or typing in the additional subjects like the balloon and chair.
> Alternatively, this refinement can be achieved using a bounding box encompassing all the subjects of interest.
> We intend to provide further clarification on this process in our revised document.

---

### Official Review · Reviewer_X5B9 · 2023-10-31

**Soundness:** 3 good
**Presentation:** 3 good
**Contribution:** 2 fair
**Rating:** 5
**Confidence:** 4

**Summary:**

This paper provides a subject repositioning task that reposition subjects within their images, which was recently introduced by Magic Editor in Google Photos (https://blog.google/products/photos/google-photos-magic-editor-pixel-io-2023/). This work combines many existing techniques, including pre-trained SAM, diffusion model, as well as image inpainting and harmonization, to build the Segment-generate-and-bLEnd (SEELE) framework for dealing with the subject repositioning task. Inspired by the textual inversion, the authors propose the task inversion for providing text prompts as input instructions of diffusion model to move or complete the subjects in images.

**Strengths:**

- This work can be considered a detailed technical document of the Magic Editor in Google Photos.
- The provided dataset ReS is helpful to evaluate the performance of subject repositioning task.

**Weaknesses:**

About the contributions:
- First, the authors claim that this work introduces the concept of subject repositioning, but obviously, it is one of the important features released by the Google Photos in May 2023, named Magic Editor in Google Photos (https://blog.google/products/photos/google-photos-magic-editor-pixel-io-2023/).
- Second, the proposed framework, namely SEELE, combines many existing techniques, and can be considered as an engineering work (like Magic Editor in Google Photos). Moreover, the authors claims that the SEELE addresses multiple generative sub-tasks in subject repositioning using a single diffusion model, on one hand, SEELE actually contains many components not only the diffusion model for generative sub-tasks, on the other hand, it is confusing what is the exact meaning of using a single diffusion model since it seems that different generative sub-tasks are tackled with different datasets at least.
- Third, about the task inversion, from Figure 3, it seems that the differences are small since it's not difficult to change the textual inversion to task inversion by using different training datasets.
- At last, ReS dataset is helpful for the evaluation of subject repositioning task, however, the comparison of this work with Magic Editor of Google Photos is not well addressed in this paper.

About the reproducibility:
- This work contains many different components with different training datasets as well as different training strategies, thus it is not easy to understand and reproduce the details.

**Questions:**

- What about the comprehensive comparison between this work and the Magic Editor in Google Photos?

---

> ### Author Response · Authors · 2023-11-11
> **Response to Reviewer X5B9 (1/2)**
>
> Dear Reviewer X5B9,
>
> Thank you for your helpful comments on our submission. Your suggestions have greatly contributed to the improvement of our paper. We have addressed each of your concerns and response in the following and are revising our submission accordingly. If you feel that we have adequately addressed your concerns, we kindly request you to consider improving the rating.
>
> Here are our responses to your insightful comments:
>
> **1.  Over-claim of introducing the concept of subject repositioning.**
>
> Thank you for your insightful comment. Initially, as mentioned in our introduction, Google Photos only provided a basic overview of the Magic Editor feature on its introductory website. However, they did not include detailed technical documentation, leaving the feature's underlying mechanisms largely unexplored. Additionally, their explanation does not cover the process flow for accomplishing the task, in contrast to our approach where we break down this complex task into several distinct sub-tasks.
> We acknowledge the need to refine our paper to more accurately represent our contribution, particularly in terms of introducing the concept of subject repositioning.
> We will revise our paper to avoid the over-claim of introducing the concept of subject repositioning.
> It's important to clarify that the sub-tasks related to subject positioning, which we detailed in our discussion, were not addressed by the Magic Editor.
>
> **2. SEELE is an engineering work and seems to adopt more than one diffusion model.**
>
> 1) This paper emphasizes that subject repositioning is more than just an amalgamation of existing techniques in engineering. Our focus is on the generative aspects, utilizing pre-trained models for non-generative tasks. We introduce innovative methods and techniques to reformulate various generative tasks within a unified framework, distinctly differentiating it from mere engineering work.
>
> 2) We underscore the use of a single diffusion model to address all generative tasks as stated in our paper. Tasks such as subject moving, completion, and local harmonization are executed using our novel task inversion technique and LoRA adapters on the same diffusion model. Meanwhile, non-generative tasks like segmentation, depth estimation, and matting are handled with pre-trained models. The stable diffusion model we employ remains unchanged (frozen) for all generative tasks, with the only adaptable parameters being the unique task tokens and LoRA parameters for each sub-task. This supports our assertion that our use of a single diffusion model is accurate and not an overstatement.
>
> 3) The utilization of diverse training datasets stems from the distinct generation requirements and training directions of our sub-tasks. Since subject repositioning represents a novel task, no single public dataset can fulfill these specific needs. Therefore, it is justified to employ different datasets for training various components, particularly the task prompts and LoRA adapters.
>
>
> **3. The task inversion is familiar with textual inversion.**
>
> Apologies for any confusion. Our discussion in our submission highlights the distinct characteristics between textual inversion and our proposed task inversion, emphasizing their differences.
>
> 1. *Differing Objectives*:
>    - Textual Inversion: This approach focuses on generating a precise description of a user-specified subject using a few images. It aims for the learned token to accurately represent the subject and adapt to various styles or scenes.
>    - Task Inversion: Our method seeks to provide a broad, image-independent instruction for the diffusion model to execute specific generative tasks. We avoid learning subject-specific concepts in the tokens, instead promoting tokens that are not tied to specific images but offer a general task guideline applicable to all images.
>
> 2. *Varied Learned Tokens:*
>    - Textual Inversion: This method utilizes one or a few learnable tokens to train on the subject, with the goal that these tokens harmonize with other text tokens.
>    - Task Inversion: In contrast, our task inversion replaces all text tokens with learned ones, focusing solely on conveying task instructions rather than providing semantic guidance.
>
> 3. *Distinct Training Approaches:*
>    - Textual Inversion: This process involves training the learned tokens with one or a few images.
>    - Task Inversion: Our approach requires a moderate training set to prevent reverting to textual inversion. It also uses a systematically constructed masking strategy to foster training in a specific generative direction.
>
> In conclusion, our task inversion represents a broader application of textual inversion, targeting different objectives, utilizing distinct training methodologies, and serving varied tasks. This has been clarified in our paper, and we will revise it for greater clarity.

---

> ### Author Response · Authors · 2023-11-11
> **Response to Reviewer X5B9 (2/2)**
>
> **4. Comparison between Magic Editor of Google Photos on ReS dataset is required.**
>
> Thank you for your insightful feedback.
> Regrettably, the Magic Editor feature of Google Photos is not available to the public at this time, and we do not have access to it.
> Therefore, our comparison is limited to the images showcased on the introductory website, as referenced in Figure 1 of our paper.
>
> **5. Reproducibility.**
>
> We apologize for any confusion caused.
> The complexity of this study arises from the challenges associated with the task of subject repositioning.
> We are committed to assisting our readers in comprehending and replicating the intricate details of our research.
> Your specific queries or points of confusion would greatly aid in clarifying any uncertainties.
>
> Regarding the dataset, our training of the SEELE model utilized only two datasets: COCO, which provides ground-truth object segmentation masks, and iHarmony4, which offers paired images for local harmonization tasks.
> These datasets, chosen for their public availability, aptly fulfill the varying requirements of different generative sub-tasks.
> Our training approach, which encompasses both subject movement and completion, employs a unified task inversion technique.
> Given that local harmonization focuses on not introducing new details in masked areas, we have modified the diffusion model to integrate the characteristics of the masked region, ensuring it aligns with the task's specific needs.
> We intend to elaborate on this in our revised submission.

---

### Official Review · Reviewer_qrCL · 2023-10-31

**Soundness:** 2 fair
**Presentation:** 3 good
**Contribution:** 2 fair
**Rating:** 5
**Confidence:** 4

**Summary:**

This paper introduces the subject repositioning task, and decouples this task into filling the void left by moving, reconstructing obscured portions, and harmonious blending. They address these subjects by the diffusion model based on task prompts. Furthermore, they propose the RES dataset for testing the performance of the model.

**Strengths:**

1.	The paper proposes the SEgment-gEnerate-and-bLEnd (SEELE) framework to solve subject repositioning problems uniformly.
2.	The paper proposes task inversion technique to generate task-specific prompts to replace text embedding in traditional Stable Diffusion.
3.	The paper proposes the RES dataset for testing the performance of the subject repositioning method.

**Weaknesses:**

1. The main concern lies in the effectiveness of the proposed task inversion method on subject repositioning. According to the quantitative comparison provided, task inversion seems to improve specific single tasks, but the improvement in subject repositioning is small (see Table 1). In the overall complex process of subject repositioning, task reversal methods may play a weak role.
2. The size of the proposed dataset may be a bit small, limiting the evaluation of the method's effectiveness.
3. It would be better to give the prompt when using SD in Table 1.

**Questions:**

Please see 'Weaknesses'.

---

> ### Author Response · Authors · 2023-11-11
> **Response to Reviewer qrCL**
>
> Dear Reviewer qrCL,
>
> We are extremely grateful for the time and effort you invested in reviewing our paper. Your constructive feedback has been instrumental in refining our research. We are making comprehensive revisions based on your valuable insights. Should our responses meet your expectations, we would appreciate your consideration for an improved rating.
>
> Here are our responses to your insightful comments:
>
> **1. The effectiveness of the proposed task inversion method.**
>
> Thank you for your insightful feedback.
> The results presented in Table 1 only partially represent the quality of the manipulated images, given that it's based on a limited sample of 200 test examples.
> This limitation is one of the reasons why we have included the detailed quantitative comparison in the appendix.
> Additionally, as you pointed out, we have extended our experiments to encompass standard inpainting and outpainting tasks, further validating the efficacy of our proposed task inversion method.
>
> Moreover, we wish to emphasize an additional benefit of the task inversion method:
> Through task inversion, we achieve seamless integration of various generative sub-tasks for subject repositioning (and potentially standard generation and manipulation tasks) using stable diffusion.
> This integration is accomplished without the need for introducing new generative models or incorporating extensive additional modules or parameters.
> This approach demonstrates the task inversion method's capacity for plug-and-play application across a diverse array of generative tasks.
>
> **2. The size of the proposed dataset may be a bit small, limiting the evaluation of the method's effectiveness.**
>
> Thank you for your comment.
> In the creation of our evaluation dataset, we prioritized diversity with great care. As detailed in our paper, the dataset comprises images sourced from over 20 distinct indoor and outdoor environments, covering a broad spectrum of subjects across more than 50 categories. This wide-ranging diversity equips us to effectively emulate real-world scenarios in open-vocabulary applications.
>
> Conversely, due to the manual nature of data collection and annotation, assembling a large-scale evaluation dataset presents significant challenges. Nevertheless, these limitations do not impact the fundamental methodology of our work. Moreover, the effectiveness and superiority of SEELE have been convincingly demonstrated within the scope of the current dataset.
>
> **3. It would be better to give the prompt when using SD in Table 1.**
>
> Sorry for the confusion and thanks for your suggestion.
> The prompt of using SD is reported in the first paragraph of Sec.3.
> Specifically, the simple prompts we used is “inpaint” and “complete the subject”, as well as complex prompts such as “Incorporate visually cohesive and high-fidelity background and texture into the provided image through inpainting” and “Complete the subject by filling in the missing region with visually cohesive and high-fidelity background and texture” for subject movement and completion tasks, respectively.
> We will highlight this in the caption of Table 1 in our revision.

---

### Author Response · Authors · 2023-11-11
**General Response**

Dear Reviewers,

We sincerely thank all the reviewers for your insightful and constructive comments. Your feedback has been invaluable in enhancing the quality and clarity of our work.
We're now working on the revision of our submission, and decide to response to your comments first, with the hope of further discussions that will enrich our work.
We appreciate the time and effort you have dedicated to reviewing our submission.

Sincerely,
Authors.

---

### Author Response · Authors · 2023-11-13
**Revision submitted**

Dear Reviewers,

We have submitted a revised version of our manuscript in response to your valuable feedback. Specifically:

i) We have renamed the generative sub-tasks to "Subject Removal," "Subject Completion," and "Subject Harmonization" to clarify their purposes.

ii) The methodology section has been restructured for a more coherent introduction of SEELE.

iii) We conducted additional ablation studies in line with your recommendations.

Please inform us of any additional concerns.

Sincerely,

The Authors.